# Subject Information Extraction for Novelty Detection with Domain Shifts

## Abstract

Unsupervised novelty detection (UND), aimed at identifying novel samples, is essential in fields like medical diagnosis, cybersecurity, and industrial quality control. Most existing UND methods assume that the training data and testing normal data originate from the same domain and only consider the distribution variation between training data and testing data. However, in real scenarios, it is common for normal testing and training data to originate from different domains, a challenge known as domain shift. The discrepancies between training and testing data often lead to incorrect classification of normal data as novel by existing methods. A typical situation is that testing normal data and training data describe the same subject, yet they differ in the background conditions. To address this problem, we introduce a novel method that separates subject information from background variation encapsulating the domain information to enhance detection performance under domain shifts. The proposed method minimizes the mutual information between the representations of the subject and background while modelling the background variation using a deep Gaussian mixture model, where the novelty detection is conducted on the subject representations solely and hence is not affected by the variation of domains. Extensive experiments demonstrate that our model generalizes effectively to unseen domains and significantly outperforms baseline methods, especially under substantial domain shifts between training and testing data.

## 1 Introduction

Novelty detection (Markou & Singh, 2003; Pimentel et al., 2014; Sabokrou et al., 2018; Pang et al., 2021) has received considerable attention for its essential applications in finance, healthcare, and security. In these fields, models must accurately predict in-distribution data and detect out-of-distribution (OOD) inputs, representing novel or unseen cases. Failing to detect such inputs can have serious consequences. OOD detection, often used interchangeably with novelty detection, is also closely related to outlier detection (Hodge & Austin, 2004), anomaly detection (Chandola et al., 2009), fault detection (Isermann, 1984), and one-class classification (Khan & Madden, 2014). For instance, unsupervised anomaly assumes most or even all training data represent normal behaviour or patterns and identifies the test data with any large deviations as anomalous. Therefore, it can be regarded as a special case of unsupervised novelty detection.

Numerous novelty detection methods (Rumelhart et al., 1986; Schölkopf et al., 1999; Breunig et al., 2000; Liu et al., 2008b; Schölkopf et al., 2001; Ruff et al., 2018; Viroli & McLachlan, 2019; Hu et al., 2020; Cai & Fan, 2022) have been proposed, For classical methods, kernel density estimation (KDE)(Parzen, 1962) utilizes a kernel function to estimate the density of data and treats the density as novelty score. OC-SVM (Schölkopf et al., 2001) tries to separate normal data from novel data by a hyperplane. Local outlier factor (LOF) (Breunig et al., 2000) regards data with lower density than its surrounding data as novel data. Autoencoder (AE) (Hinton & Salakhutdinov, 2006) uses reconstruction error as a novelty metric. Isolation forest (IF) (Liu et al., 2008a) uses the length of iTree to detect novel samples. As for recent state-of-the-art methods, ALAD (Zenati et al., 2018) based on bi-directional GANs, uses reconstruction errors based on these adversarially learned features to determine if a data sample is novel. (Ruff et al., 2018) released DeepSVDD, which utilizes a neural network to enclose the representations of normal data in a hypersphere in the latent space with minimal volume. MO-GAAL (Liu et al., 2019) can directly generate informative potential

outliers based on the mini-max game between a generator and a discriminator and n generate a reference distribution for the whole dataset to provide sufficient information to assist the classifier in describing a boundary that can separate novel samples from normal data effectively. DROCC (Goyal et al., 2020) which is on the basis of low-dimensional manifold assumption on normal data, generates negative samples to provide general and robust identification on novel samples. SUOD (Zhao et al., 2021), which is a comprehensive acceleration framework for novelty (outlier) detection, generates random low-dimensional subspace for base models and uses the output of unsupervised models as pseudo ground truth. PLAD (Cai & Fan, 2022) which is based on perturbation learning, learns small perturbations to perturb normal data and learns a classifier to classify the normal data and the perturbed data into two different classes. Then, data classified as perturbed is considered to be novel (anomalous). DIF (Xu et al., 2023), a deep-learning version of isolation forest (Liu et al., 2008b), enables non-linear partition on subspaces of varying sizes, offering a more effective novelty (anomaly) isolation solution.

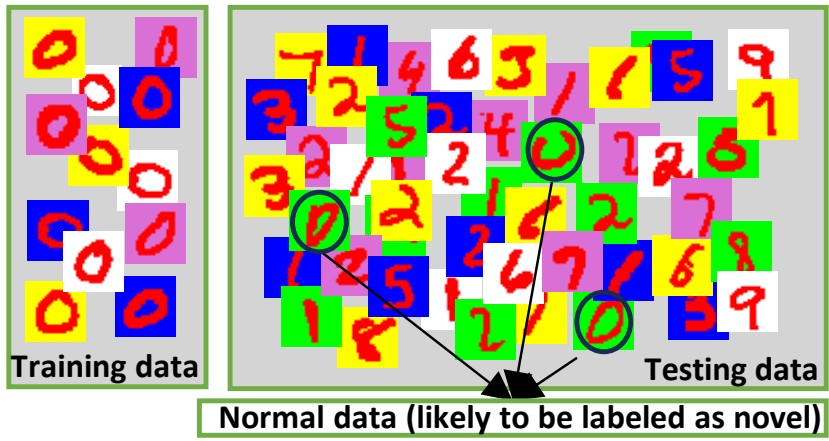

Figure 1: An illustration of the novelty detection task. The training data (left) consists of images of the digit '0' presented in four backgrounds. The testing data (right) includes images of multiple digits (0-9) in seen backgrounds and entirely unseen backgrounds. Although the '0' digits in the test set are normal, some of them are likely to be labelled as novel due to the shift in background.

A significant limitation of many existing unsupervised novelty detection (UND) methods is that, while they acknowledge the difference in distributions between training and test data, they often assume that both training and normal test data originate from the same domain. However, in real-world scenarios, it is common for normal test data and training data to be sourced from different domains, a challenge referred to as domain shift. For example, training and normal test data may be collected under varying environmental conditions, from different individuals, or at different times. A typical instance of this issue arises when training and test data describe the same subject but differ in background conditions, as illustrated in Figure 1. Such domain shifts can significantly affect model performance by leading to misclassifications, particularly when the domain differences in the test data are not accounted for during training (Wu et al., 2023).

To address the challenges of domain shift, methods such as domain generalization, empirical risk minimization(ERM) (Vapnik, 1991) and invariant risk minimization (IRM)(Arjovsky et al., 2019) have been developed. These approaches generally require task-specific labels (e.g., classification labels) and domain labels to address domain differences in the training data. In some instances, hybrid or auxiliary labels are necessary to further improve model performance. However, labelling domain-specific information for each data point can be resource-intensive. In contrast, our proposed model simplifies this process by only requiring the number of domains from which the data is sourced, reducing the labelling burden.

To address the challenge of background (domain) shifts between training and normal test data in unsupervised novelty detection, we propose a novel and effective method called Subject-Novelty Detection (SND). SND disentangles subject information from background features in the training

data, allowing the model to focus on subject-specific features during novelty detection. This enables SND to maintain strong performance even when the normal test data exhibits entirely different background characteristics from the training data. Unlike other domain adaptation methods, which often require both task and domain labels for each data point, SND only necessitates knowledge of the number of domains, making it more efficient while preserving high accuracy in novelty detection. Our main contributions are as follows:

- We introduce Subject-Novelty Detection (SND), which isolates subject information from background variations, enabling robust detection even under significant domain shifts.

- SND eliminates the need for prior knowledge of the subject or background details. It only requires information on the number of domains in the training data, making it efficient and adaptable for detecting novelty in domain-shift scenarios.

- We extensively compare SND with existing methods and domain shift techniques, demonstrating that SND achieves state-of-the-art results in various scenarios.

## 2 RELATED WORK

### 2.1 UNSUPERVISED NOVELTY DETECTION

Novelty detection (ND) is usually an unsupervised learning problem, where training data are unlabeled and most or all of them are normal data. Novelty detection can be divided into two types. The first type is to identify novel samples in a dataset by training a machine learning model, where the novel samples or outliers are identified once the model training is finished. The methods of this type are often based on density estimation or use some robust loss functions. The typical methods include robust kernel density estimation, Gaussian mixture models, robust PCA (Xu et al., 2010; Candès et al., 2011), low-rank representations (Liu et al., 2012), robust kernel PCA (Fan & Chow, 2019), etc. The second type is to train a model on a training dataset without any outliers or with a very small fraction of unlabeled outliers. This setting is the same as unsupervised anomaly detection and one-class classification. Typical methods include PCA, autoencoder (Rumelhart et al., 1986), LOF (Breunig et al., 2000), Isolation forest (Liu et al., 2008b), OC-SVM (Schölkopf et al., 2001), SVDD (Tax & Duin, 2004), Deep SVDD (Ruff et al., 2018), DAGMM (Viroli & McLachlan, 2019), AnoGAN (Schlegl et al., 2017), HRN (Hu et al., 2020), PLAD (Cai & Fan, 2022), DPAD (Fu et al., 2024), etc. In this study, we focus on the second type.

### 2.2 DOMAIN ADAPTATION AND TRANSFER LEARNING

Domain adaptation and transfer learning strategies play a pivotal role in enhancing the performance of learning models when faced with new tasks or domains. Traditional machine learning models are often trained on specific datasets, but real-world scenarios frequently present data distributions that vary across tasks or domains. Domain adaptation techniques address discrepancies between data distributions in source and target domains. These include instance re-weighting, feature mapping, and adversarial learning (Tzeng et al., 2017). Transfer learning leverages knowledge from related tasks to mitigate the data and computational requirements of new tasks, finding success in computer vision and natural language processing domains (Pan & Yang, 2009).

Recent advances in domain adaptation and transfer learning include unsupervised domain adaptation techniques that align source and target domain features without requiring target domain labels (Tzeng et al., 2017). Multi-source domain adaptation improves model performance by integrating data from multiple source domains (Zhao et al., 2018). Cross-modal transfer learning has made strides in knowledge transfer between different modalities (Chen et al., 2019). Meta-learning techniques, such as Model-Agnostic Meta-Learning (MAML), excel in rapid adaptation to new tasks (Finn et al., 2017). Self-supervised learning reduces the need for labelled data in transfer learning scenarios (Chen et al., 2020). More recently, some researchers explored OOD detection combined with domain adaptation (Oza et al., 2020; Yang et al., 2023; Carvalho et al., 2024), focusing primarily on transitioning from one scene to another.

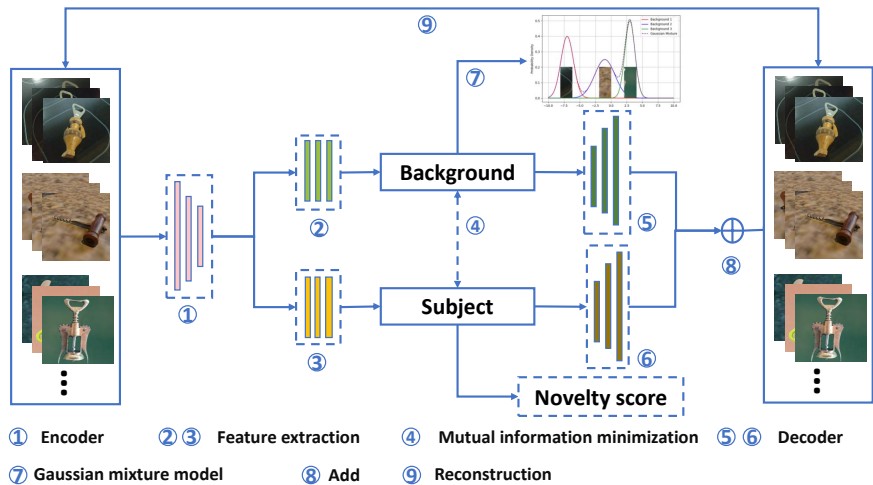

Figure 2: An overview of the proposed SND model.

# 3 UNSUPERVISED SUBJECT NOVELTY DETECTION

## 3.1 PROBLEM FORMULATION

To be precise, suppose we have a training dataset consisting of $N$ images, denoted as $\mathcal{D} = \{\mathbf{x}_1, \mathbf{x}_2, \ldots, \mathbf{x}_N\}$, in which each $\mathbf{x}_i \in \mathbb{R}^{C \times H \times W}$ has a background $b_i$ chosen from a set of $K$ different backgrounds $\mathcal{B} = \{B_1, B_2, \ldots, B_K\}$ and all or most of the $N$ samples are normal. Notably, although the total number of backgrounds $K$ is known, the specific background type of each image is unknown. This setting is practical since data or images collected often come from different backgrounds (or domains more generally) and labelling the backgrounds is costly. We consider a test set $\mathcal{D}' = \{\mathbf{x}_1', \mathbf{x}_2', \ldots, \mathbf{x}_M'\}$, where the background $b_i'$ of each $\mathbf{x}_i'$ is chosen from a larger set $\tilde{\mathcal{B}} = \{B_1, B_2, \ldots, B_K, B_{K+1}, \ldots, B_{K+K'}\}$. Note that $B_{K+1}, \ldots, B_{K+K'}$ are actually new backgrounds different from $B_1, B_2, \ldots, B_K$ and $K'$ is unknown. Our goal is to learn a model from $\mathcal{D}$ to determine whether a new sample from $\mathcal{D}'$ is a novel sample in terms of the subject information rather than the background information. This is a nontrivial task because the domain of normal data changed, or in other words, the distribution of normal data changed.

A simple example of the task is shown in Figure 1, where the training set contains images of digit $'0'$ with $4$ different coloured backgrounds (white, yellow, blue, pink), and the testing set contains images of digits $(0-9)$ with $5$ different coloured backgrounds (white, yellow, blue, pink, green). To evaluate the performance of methods under extreme background shifts, the background of digit $'0'$ (testing normal data) is set to a completely unseen green background, while backgrounds of digits $(1-9)$ (testing novel data) are set to all $5$ colours. Our aim is to identify digits $(1-9)$ as novel samples which contain different subject information while treating digits $0$ as normal samples which differ from training data only in background information.

Classical ND tasks only consider the distribution difference between the training data and testing novel data. Our tasks consider not only the distribution difference mentioned before but also the background (domain) shift between training data and testing normal data, which leads to distribution difference between them. Thus unsupervised subject novelty detection is a more complicated novelty detection task. Classical ND methods have high false positive rates on this task because they will label the normal samples with new backgrounds as novel samples.

## 3.2 PROPOSED MODEL

We aim to address the challenge of isolating subject information from varying backgrounds for improved novelty detection. One key point is to learn representations that separate subject and back-

ground features in an unsupervised manner, allowing the model to detect novel subject information despite shifts in background domains. The process of our method is illustrated in Figure 2.

The process begins with a feature extraction network $G_{\theta_f} : \mathbb{R}^{C \times H \times W} \to \mathbb{R}^d$ with parameters $\theta_f$, which processes the input image $\mathbf{x}$ and generates a feature representation $\mathbf{z}_f$, i.e.,

$$\mathbf{z}_f = G_{\theta_f}(\mathbf{x}). \tag{1}$$

This representation is then decomposed into two distinct components, a subject feature $\mathbf{z}_s$ and a background feature $\mathbf{z}_b$, through two neural networks $F_{\theta_s} : \mathbb{R}^d \to \mathbb{R}^d$ and $F_{\theta_b} : \mathbb{R}^d \to \mathbb{R}^d$, i.e.,

$$\mathbf{z}_s = F_{\theta_s}(\mathbf{z}_f), \quad \mathbf{z}_b = F_{\theta_b}(\mathbf{z}_f). \tag{2}$$

It is nontrivial to guarantee that $\mathbf{z}_s$ and $\mathbf{z}_b$ exclusively represent the subject and background information, respectively. With the insights provided by (Cheng et al., 2020), we propose to minimize the mutual information $I(\mathbf{z}_s; \mathbf{z}_b)$ between $\mathbf{z}_s$ and $\mathbf{z}_b$, which will encourage the two parts to be statistically independent. The mutual information is estimated using a neural network $\xi_{\theta_m}$ based on the following formulation

$$\hat{I}_{\mathrm{MI}}(\mathbf{z}_s; \mathbf{z}_b) = \frac{1}{N} \sum_{i=1}^{N} \left[ \log \xi_{\theta_m}(\mathbf{z}_b^{(i)} | \mathbf{z}_s^{(i)}) - \frac{1}{N} \sum_{j=1}^{N} \log \xi_{\theta_m}(\mathbf{z}_b^{(j)} | \mathbf{z}_s^{(i)}) \right]. \tag{3}$$

The full derivation of the mutual information estimation is detailed in Appendix A. It is worth noting that making $\mathbf{z}_s$ and $\mathbf{z}_b$ independent cannot ensure that $\mathbf{z}_s$ is composed of the subject information and $\mathbf{z}_b$ is composed of the background information. $\mathbf{z}_s$ may represent background information while $\mathbf{z}_b$ may represent subject information. In other words, we cannot identify their correspondences.

Fortunately, by assuming that the number of background types is $K$ and $K$ is different from the number of potential clusters in the subject information, we can distinguish between subject information and background information. Specifically, inspired by (Zong et al., 2018), we use a deep Gaussian Mixture Model (GMM) with $K$ components to model $\mathbf{z}_b$. $S_{\theta_g} : \mathbb{R}^d \to \mathbb{R}^K$ is a neural network projecting $\mathbf{z}_b$ to $\hat{\gamma}_k^i$, which represents the soft membership prediction for each mixture component.

$$\hat{\gamma}^i = \mathrm{softmax}(S_{\theta_g}(\mathbf{z}_b^i)) \tag{4}$$

The modelling will encourage $\mathbf{z}_b$ to capture $K$ clusters, making it different from the subject information. Denoting $\hat{\pi}_k$ the weight of the $k$-th Gaussian component, $\hat{\boldsymbol{\mu}}_k \in \mathbb{R}^d$ the mean, and $\hat{\boldsymbol{\Sigma}}_k \in \mathbb{R}^{d \times d}$ the covariance matrix of the $k$-th component.

$$\hat{\pi}_k = \frac{1}{L} \sum_{i=t_1}^{t_L} \hat{\gamma}_k^i, \quad \hat{\mu}_k = \frac{\sum_{i=t_1}^{t_L} \hat{\gamma}_k^i \mathbf{z}_b^i}{\sum_{i=t_1}^{t_L} \hat{\gamma}_k^i}, \quad \hat{\Sigma}_k = \frac{\sum_{i=t_1}^{t_L} \hat{\gamma}_k^i (\mathbf{z}_b^i - \hat{\mu}_k)(\mathbf{z}_b^i - \hat{\mu}_k)^\top}{\sum_{i=t_1}^{t_L} \hat{\gamma}_k^i} \tag{5}$$

Given a randomly sampled batch of data $\{\mathbf{x}^i\}_{i=t_1}^{t_L} \subseteq \mathcal{D}$, $\{t_1, \ldots, t_L\} \subseteq \{1, \ldots, N\}$ and their background feature vectors $\{\mathbf{z}_b^i\}_{i=t_1}^{t_L}$ with batch size $L$, we define the following background energy function

$$E(\mathbf{z}_b^i) = -\log \left( \sum_{k=1}^{K} \hat{\pi}_k (2\pi)^{-d/2} |\hat{\boldsymbol{\Sigma}}_k|^{-1/2} \exp \left( -\frac{1}{2} (\mathbf{z}_b^i - \hat{\boldsymbol{\mu}}_k)^\top \hat{\boldsymbol{\Sigma}}_k^{-1} (\mathbf{z}_b^i - \hat{\boldsymbol{\mu}}_k) \right) \right) \tag{6}$$

The identification of $\mathbf{z}_b$ together with its independence to $\mathbf{z}_s$ sure that $\mathbf{z}_s$ captures the subject information naturally.

Nevertheless, we still need to ensure that $\mathbf{z}_s$ and $\mathbf{z}_b$ preserve the original information of the input $\mathbf{x}$. This can be done by letting them be able to reconstruct the input $\mathbf{x}$. Specifically, we feed $\mathbf{z}_s$ and $\mathbf{z}_b$ into two different decoders $H_{\theta_s'} : \mathbb{R}^d \to \mathbb{R}^{C \times H \times W}$ and $H_{\theta_b'} : \mathbb{R}^d \to \mathbb{R}^{C \times H \times W}$, i.e.,

$$\mathbf{x}_s = H_{\theta_s'}(\mathbf{z}_s), \quad \mathbf{x}_b = H_{\theta_b'}(\mathbf{z}_b), \tag{7}$$

and summarize their outputs as the reconstruction for $\mathbf{x}$, i.e.,

$$\hat{\mathbf{x}} = \mathbf{x}_s + \mathbf{x}_b. \tag{8}$$

By isolating subject and background information, our method can focus on detecting novelty in the subject information, even when there is significant variation in the background. This feature decomposition and reconstruction mechanism ensures robustness to background changes and facilitates accurate novelty detection.

### 3.3 Training and Evaluation

Here, we summarize the entire process of the proposed method. Due to the mutual information estimation and GMM parts, we can ensure that both $\mathbf{z}_s$ and $\mathbf{z}_b$ contain necessary information about the subject and background respectively, without worrying that one has learned most of the information while the other has not learned anything. The loss function $L_{\text{rec}}(\mathbf{x}, \hat{\mathbf{x}})$ represents the reconstruction error between the original image $\mathbf{x}$ and the reconstructed output image $\hat{\mathbf{x}}$, which is expressed as

$$L_{\text{rec}}(\mathbf{x}, \hat{\mathbf{x}}) = \|\mathbf{x} - \hat{\mathbf{x}}\|_2^2. \tag{9}$$

We calculate the weighted sum of all the loss terms to obtain the total loss for the proposed model:

$$L_{\text{total}} = L_{\text{rec}}(\mathbf{x}, \mathbf{x_r}) + \omega_1 E(\mathbf{z}_b) + \omega_2 \hat{I}_{\text{MI}}(\mathbf{z}_s; \mathbf{z}_b), \tag{10}$$

where $\omega_1$ and $\omega_2$ are non-negative hyperparameters and the parameters to learn are $\{\theta_f, \theta_s, \theta_b, \theta_m, \theta_g, \theta'_s, \theta'_b\}$

After our model is well-trained, we can use Kernel Density Estimation (KDE) which is a simple yet effective method to conduct novelty detection. Specifically, we denote the subject feature vectors of the training set as $\mathcal{D}_s = \{\mathbf{z}_s^{(1)}, \mathbf{z}_s^{(2)}, \ldots, \mathbf{z}_s^{(N)}\} = \{F_{\theta_s}(G_{\theta_f}(\mathbf{x})) : \mathbf{x} \in \mathcal{D}\}$. Given a test sample $\mathbf{x}_{\text{new}}$, its subject feature vector is $\mathbf{z}_s^{\text{new}} = F_{\theta_s}(G_{\theta_f}(\mathbf{x}_{\text{new}}))$. Thus, the novelty score (NS) of $\mathbf{x}_{\text{new}}$ is given by the negative density of $\mathbf{z}_s^{\text{new}}$, i.e.,

$$\text{NS}(\mathbf{x}_{\text{new}}) = -\hat{p}(\mathbf{z}_s^{\text{new}}) = -\frac{1}{n(2\pi h^2)^{d/2}} \sum_{i=1}^{n} \exp\left(-\frac{\|\mathbf{z}_s^{\text{new}} - \mathbf{z}_s^{(i)}\|^2}{2h^2}\right) \tag{11}$$

where $h$ is the bandwidth parameter controlling the smoothness of the estimated density, and $d$ is the dimensionality of $\mathbf{z}_s$. A higher novelty score $\text{NS}(\mathbf{x}_{\text{new}})$ indicates that the subject of $\mathbf{x}_{\text{new}}$ has a lower likelihood of belonging to the distribution of subjects in the training data $D$.

In general, the proposed method learns comprehensive subject features $\mathbf{z}_s$ and background features $\mathbf{z}_b$ using our objective function defined in equation 10, which includes the weighted sum of reconstruction loss, energy of $\mathbf{z}_b$, and mutual information between $\mathbf{z}_s$ and $\mathbf{z}_b$. For novelty detection, KDE fitted on the training set is applied to the subject feature $\mathbf{z}_s$ of the test sample, and the novelty score for a test sample is determined using equation 11.

## 4 Experiments

In this section, we benchmark various methods using numerical experiments on several challenging and widely used datasets. To evaluate performance, we selected 9 out of 10 classes from multiple scenarios as normal classes in the training set where the model is trained on these 9 classes. For testing, images from a different unseen background are used.

### 4.1 Implementation Details and Datasets

We evaluated the proposed method on three challenging datasets: Multi-background MNIST, Multi-background Fashion-MNIST, and Kurcuma. To address the limitations in variability in the original MNIST and Fashion-MNIST datasets (LeCun et al., 1998; Xiao et al., 2017), we introduced domain shifts by altering background colours. For the Multi-background MNIST dataset, the model was trained using 'blue', 'yellow', and 'white' backgrounds and tested on a previously unseen 'green' background. Similarly, for the Multi-background Fashion-MNIST, the model was trained on 'blue', 'green', 'purple', and 'white' backgrounds and evaluated on a new 'yellow' background. These setups evaluated the model's generalization to unseen domains.

Additionally, the Kurcuma dataset, containing diverse real-world images, was used to further test the model's adaptability across synthetic and real-world scenarios. Detailed descriptions of the datasets and additional results are provided in Appendices E and D.

**Evaluation Methods and Metrics**

We conducted an extensive performance evaluation by comparing our model against a wide range of recent state-of-the-art novelty detection methods. It is worth noting that classical methods perform poorly when dealing with complex scenarios and high-dimensional data in this task, we include no classical methods in our baselines. Methods compared includes AnoGAN (Schlegl et al.,

Table 1: Average AUROCs (%) in novelty detection on Multi-background MNIST. In each case, the best result is marked in bold.

| Method | 0 | 1 | 2 | 3 | 4 | 5 | 6 | 7 | 8 | 9 | Average |
|---|---|---|---|---|---|---|---|---|---|---|---|
| COPOD | 62.82 | 70.33 | 63.77 | 64.41 | 65.80 | 64.33 | 64.44 | 66.05 | 63.81 | 65.95 | 65.17 |
| SUOD | 64.52 | 67.42 | 65.79 | 67.72 | 70.17 | 69.37 | 65.46 | 67.24 | 65.10 | 67.70 | 67.05 |
| MO_GAAL | 61.41 | 72.20 | 69.40 | 77.73 | 65.74 | 58.00 | 71.48 | 71.23 | 73.13 | 73.63 | 69.40 |
| DeepSVDD | 63.92 | 58.84 | **72.46** | 56.69 | 48.24 | **88.63** | 66.82 | 76.00 | 62.02 | 62.95 | 65.66 |
| ALAD | 27.97 | 29.07 | 7.58 | 17.22 | 8.85 | 25.06 | 13.84 | 9.77 | 22.37 | 16.93 | 17.87 |
| ECOD | 57.29 | 61.46 | 60.37 | 60.41 | 62.13 | 61.19 | 60.22 | 61.29 | 59.67 | 61.53 | 60.56 |
| INNE | 61.23 | 57.83 | 63.72 | 63.15 | 61.80 | 66.64 | 62.72 | 65.09 | 58.84 | 61.37 | 62.24 |
| AnoGAN | 4.86 | 0.36 | 32.28 | 52.98 | 52.43 | 43.97 | 9.63 | 33.07 | 22.85 | 36.38 | 28.88 |
| ERM | 36.42 | 95.36 | 42.00 | 42.25 | 38.25 | 40.29 | 51.65 | 51.96 | 48.00 | 40.54 | 48.67 |
| IRM | 35.65 | 96.32 | 40.41 | 37.54 | 38.47 | 37.09 | 47.72 | 63.56 | 47.82 | 41.29 | 48.59 |
| GNL | 61.47 | 93.07 | 50.04 | 82.73 | 63.20 | 54.30 | 60.23 | 68.56 | 60.58 | 56.51 | 65.07 |
| SND | **85.74** | **97.68** | 71.35 | **84.40** | **75.55** | 74.59 | **90.39** | **85.09** | **80.24** | **74.08** | **82.27** |

2017), DeepSVDD (Ruff et al., 2018), XGBOD (Zhao & Hryniewicki, 2018), ALAD (Zenati et al., 2018),INNE (Bandaragoda et al., 2018), MO-GAAL(Liu et al., 2019), COPOD (Li et al., 2020), ROD (Almardeny et al., 2020), SUOD (Zhao et al., 2021), and ECOD (Li et al., 2022). The hyperparameters for the methods listed above were set according to the default settings provided by the PyOD(Zhao et al., 2019).

Table 2: Average AUPRCs (%) in novelty detection on Multi-background MNIST. In each case, the best result is marked in bold.

| Method | 0 | 1 | 2 | 3 | 4 | 5 | 6 | 7 | 8 | 9 | Average |
|---|---|---|---|---|---|---|---|---|---|---|---|
| COPOD | 22.78 | 27.06 | 23.24 | 23.56 | 24.28 | 23.51 | 23.57 | 24.41 | 23.27 | 24.36 | 24.00 |
| SUOD | 23.37 | 24.93 | 24.22 | 25.21 | 26.93 | 26.31 | 23.99 | 24.93 | 23.79 | 25.30 | 24.90 |
| MO_GAAL | 28.55 | 35.31 | 35.44 | 38.52 | 29.82 | 28.87 | 37.22 | 32.62 | 35.99 | 35.87 | 33.82 |
| DeepSVDD | 26.82 | 22.42 | 43.75 | 23.85 | 25.15 | 67.47 | 31.67 | 37.81 | 41.87 | 30.85 | 35.17 |
| ALAD | 13.95 | 14.02 | 11.36 | 12.64 | 11.53 | 16.20 | 12.05 | 11.68 | 13.51 | 12.75 | 12.97 |
| ECOD | 20.47 | 22.11 | 21.64 | 21.66 | 22.41 | 22.01 | 21.61 | 21.38 | 22.04 | 22.15 | 21.75 |
| INNE | 24.32 | 23.76 | 26.23 | 25.97 | 25.81 | 27.78 | 25.75 | 27.13 | 24.26 | 24.82 | 25.58 |
| AnoGAN | 13.94 | 13.72 | 21.56 | 25.21 | 25.04 | 21.62 | 14.73 | 22.03 | 16.68 | 22.23 | 19.68 |
| ERM | 86.85 | 99.39 | 88.60 | 87.99 | 88.18 | 88.79 | 90.59 | 90.01 | 90.06 | 88.24 | 89.87 |
| IRM | 86.31 | 99.53 | 88.37 | 86.83 | 87.81 | 88.11 | 90.13 | 93.15 | 90.02 | 88.33 | 89.86 |
| GNL | 93.92 | 99.02 | 91.39 | 96.96 | 94.22 | 92.88 | 93.04 | 94.87 | 93.94 | 91.83 | 94.21 |
| SND | **97.49** | **99.73** | **95.30** | **97.56** | **96.40** | **94.42** | **98.69** | **97.15** | **97.24** | **96.16** | **97.01** |

Furthermore, we evaluated our approach against GNL, a recently proposed method for novelty detection across domain transformations (Cao et al., 2023). The hyperparameters for GNL were set following the recommendations from the original publication. We also compared our method with two key domain adaptation techniques, ERM and IRM, both followed by a KDE step for novelty detection. This allowed us to evaluate our model's effectiveness in handling domain shifts and identifying novel data in unseen environments.

We employed two common metrics to evaluate the performance of novelty detection: (i) Area Under the Receiver Operating Characteristic curve (AUROC), which can be interpreted as the probability that a positive sample has a higher discriminative score than a negative sample; and (ii) Area Under the Precision-Recall curve (AUPRC), an ideal metric for adjusting extreme differences between positive and negative base rates.

Table 3: Average AUROCs (%) in novelty detection on Multi-background Fashion-MNIST. In each case, the best result is marked in bold.

| Method | 0 | 1 | 2 | 3 | 4 | 5 | 6 | 7 | 8 | 9 | Average |
|--------|---|---|---|---|---|---|---|---|---|---|---------|
| COPOD | 59.93 | 65.17 | 58.80 | 62.42 | 59.32 | 58.36 | 58.85 | 63.18 | 55.95 | 59.11 | 60.11 |
| SUOD | 63.23 | 65.37 | 61.04 | 65.43 | 62.08 | 62.79 | 61.18 | 63.30 | 61.60 | 62.97 | 62.90 |
| MO_GAAL | 47.03 | 56.49 | 44.29 | 58.47 | 54.61 | 65.68 | 46.14 | 59.91 | 48.92 | 49.55 | 53.11 |
| DeepSVDD | 64.05 | 62.09 | 57.48 | 60.00 | **68.68** | 63.19 | 66.59 | 52.20 | 61.33 | 70.04 | 62.57 |
| ALAD | 39.54 | 31.90 | 35.54 | 25.79 | 29.30 | 21.12 | 31.76 | 15.40 | 36.37 | 34.10 | 30.08 |
| ECOD | 54.68 | 57.48 | 51.80 | 54.32 | 52.77 | 55.27 | 53.04 | 57.36 | 54.66 | 55.39 | 54.68 |
| INNE | 64.49 | 59.65 | 64.74 | 65.03 | 66.05 | 64.52 | 65.47 | 59.24 | **69.77** | 67.53 | 64.65 |
| AnoGAN | 85.23 | 97.20 | 57.19 | 87.02 | 53.07 | 56.32 | 40.99 | 48.52 | 67.14 | **78.11** | 67.08 |
| ERM | 58.61 | 32.47 | 56.45 | 45.37 | 47.14 | 90.60 | 53.76 | 39.74 | 55.78 | 36.16 | 51.61 |
| IRM | 57.80 | 32.01 | 56.92 | 43.71 | 48.29 | 89.72 | 52.01 | 37.21 | 56.09 | 30.12 | 50.39 |
| GNL | 63.31 | 88.55 | 43.68 | 81.34 | 57.19 | 77.82 | 43.47 | 85.12 | 40.00 | 72.69 | 65.32 |
| SND | **89.03** | **93.21** | **70.36** | **87.75** | 65.34 | **84.16** | **78.03** | **90.73** | 62.36 | 77.64 | **79.86** |

## 4.2 RESULTS AND DISCUSSION

In this section, we evaluate and analyze the performance of our method compared to recent state-of-the-art novelty detection baseline methods across the mentioned datasets. SND consistently demonstrates superior performance in the extensive experiments.

To further demonstrate the model's robustness and generalization capabilities, we present results from experiments on varying background colours and different numbers of backgrounds in the training dataset. These experiments allowed us to evaluate the model's performance under domain shifts with previously unseen backgrounds. The main text provides results for these two specific scenarios, while results for additional experiments with different background settings are included in Appendix F, Appendix G, and Appendix H for reference.

Table 4: Average AUPRCs (%) in novelty detection on Multi-background Fashion-MNIST. In each case, the best result is marked in bold.

| Method | 0 | 1 | 2 | 3 | 4 | 5 | 6 | 7 | 8 | 9 | Average |
|--------|---|---|---|---|---|---|---|---|---|---|---------|
| COPOD | 21.98 | 23.84 | 21.57 | 22.32 | 21.61 | 20.71 | 21.61 | 22.95 | 20.81 | 20.91 | 21.83 |
| SUOD | 24.03 | 23.80 | 23.46 | 24.11 | 23.53 | 22.62 | 23.49 | 22.81 | 23.83 | 22.79 | 23.45 |
| MO_GAAL | 20.10 | 26.41 | 20.72 | 32.13 | 26.82 | 28.55 | 18.11 | 26.98 | 20.40 | 21.79 | 24.20 |
| DeepSVDD | 27.54 | 25.60 | 24.12 | 24.13 | 30.16 | 27.19 | 29.25 | 24.63 | 27.70 | 33.01 | 27.33 |
| ALAD | 17.57 | 27.77 | 15.59 | 13.23 | 13.98 | 12.93 | 14.75 | 12.25 | 16.06 | 17.54 | 16.17 |
| ECOD | 19.97 | 20.43 | 19.17 | 19.24 | 19.33 | 19.60 | 19.53 | 20.46 | 20.43 | 19.58 | 19.77 |
| INNE | 25.99 | 23.34 | 26.04 | 26.36 | 26.50 | 25.46 | 26.56 | 24.11 | 28.62 | 27.21 | 26.02 |
| AnoGAN | 53.83 | 78.55 | 23.23 | 52.45 | 20.90 | 28.95 | 16.18 | 47.45 | 38.88 | 49.85 | 41.03 |
| ERM | 94.18 | 84.19 | 93.39 | 88.79 | 91.49 | 97.84 | 92.90 | 85.79 | 92.16 | 86.01 | 90.68 |
| IRM | 94.02 | 83.99 | 93.15 | 88.51 | 92.07 | 97.65 | 92.38 | 85.29 | 92.23 | 83.92 | 90.32 |
| GNL | 94.64 | **98.50** | 91.07 | 97.64 | 93.46 | 97.25 | 89.80 | 98.39 | 85.35 | 95.84 | 94.19 |
| SND | **98.12** | 95.72 | **95.46** | **98.53** | **93.62** | **98.63** | **96.36** | **98.79** | **92.98** | **97.13** | **96.53** |

In Table 1, we compare the performance of various methods on novelty detection using the Multi-background MNIST dataset, focusing on AUROC scores. Our proposed method, SND, achieves the highest average AUROC of 82.27%, outperforming baseline methods like COPOD (65.17%) and SUOD (67.05%). ERM and IRM, two domain adaptation techniques followed by KDE for novelty detection, perform significantly lower with averages of 48.67% and 48.59%, respectively. Notably, SND excels in digits such as 0 (85.74%) and 1 (97.68%), demonstrating superior generalization across different digits.

Table 2 shows that SND also leads in AUPRC scores with an average of 97.01%. This is significantly higher than GNL (94.21%) and ERM (89.87%). The performance of SND is consistent across all digits, particularly in digits like 1 (99.73%) and 7 (97.15%), confirming its robustness in novelty

Table 5: Average AUROC (%) for Novelty Detection on the Kurcuma dataset using data from seven different scenarios as test sets.

| Method | 0 | 1 | 2 | 3 | 4 | 5 | 6 | 7 | 8 | Average |
|--------|---|---|---|---|---|---|---|---|---|---------|
| ALAD | 47.60 | 46.97 | 48.78 | 55.30 | 47.37 | 49.41 | 48.87 | 50.12 | 50.27 | 49.41 |
| COPOD | 45.51 | 43.85 | 55.37 | 50.76 | 50.99 | 48.32 | 54.48 | 47.18 | 51.75 | 49.80 |
| DeepSVDD | 48.79 | 46.12 | 49.64 | 50.87 | 51.27 | 48.31 | 52.12 | 50.55 | 50.22 | 49.77 |
| ECOD | 45.97 | 43.95 | 55.23 | 50.63 | 49.10 | 49.31 | 54.56 | 47.37 | 49.52 | 49.51 |
| INNE | 47.09 | 41.52 | 58.66 | 53.47 | 45.48 | 42.89 | 59.00 | 47.60 | 52.87 | 49.84 |
| AnoGAN | 47.39 | 52.70 | 47.23 | 50.48 | 57.89 | 47.59 | 49.42 | 46.73 | 48.82 | 49.81 |
| ERM | 55.37 | 45.96 | 47.05 | 50.42 | 51.17 | 48.36 | 47.08 | 55.49 | 50.18 | 50.12 |
| IRM | 53.71 | 47.71 | 47.23 | 50.19 | 51.90 | 48.76 | 50.25 | 53.45 | 43.56 | 49.64 |
| GNL | 49.29 | 41.46 | **77.32** | 65.75 | 48.81 | **62.32** | **71.97** | 61.11 | **71.67** | 61.08 |
| SND | **72.70** | **71.56** | 74.13 | **65.85** | **78.67** | 59.98 | 64.18 | **71.72** | 70.85 | **69.96** |

detection tasks. This table highlights the effectiveness of SND in detecting novel examples even in unseen domains.

In Table 3, which analyzes novelty detection on the **Multi-background Fashion-MNIST** dataset, our method, SND, consistently achieves superior performance, with an average AUROC of 79.86%. SND excels in several classes, particularly class 0 (89.03%) and class 1 (93.21%), outperforming other methods such as GNL (65.32%) and DeepSVDD (62.57%). Both ERM and IRM show significantly lower average AUROCs of 51.61% and 50.39%, respectively, indicating their reduced effectiveness.

Table 6: Average AUPRC (%) for Novelty Detection on the Kurcuma dataset using data from seven different scenarios as test sets.

| Method | 0 | 1 | 2 | 3 | 4 | 5 | 6 | 7 | 8 | Average |
|--------|---|---|---|---|---|---|---|---|---|---------|
| ALAD | 91.82 | 94.30 | 90.33 | 80.77 | 95.29 | 82.79 | 78.94 | 93.48 | 93.08 | 88.98 |
| COPOD | 91.31 | 94.45 | 91.74 | 79.00 | 94.96 | 82.72 | 81.30 | 93.22 | 93.24 | 89.10 |
| DeepSVDD | 92.14 | 94.15 | 90.22 | 79.04 | 95.52 | 83.07 | 81.11 | 93.72 | 92.96 | 89.10 |
| ECOD | 91.55 | 94.26 | 91.52 | 79.04 | 94.92 | 83.30 | 80.90 | 93.18 | 93.19 | 89.10 |
| INNE | 91.80 | 93.80 | 92.77 | 80.09 | 94.74 | 79.96 | 83.45 | 93.29 | 93.90 | 89.31 |
| AnoGAN | 92.36 | 95.32 | 89.04 | 79.28 | 96.49 | 82.30 | 80.20 | 93.05 | 93.29 | 89.04 |
| ERM | 93.41 | 94.52 | 89.41 | 79.38 | 95.86 | 82.08 | 77.95 | 94.62 | 92.93 | 88.91 |
| IRM | 93.45 | 94.79 | 88.99 | 79.95 | 95.81 | 83.22 | 77.84 | 94.42 | 92.96 | 89.05 |
| GNL | 91.88 | 93.97 | **96.47** | 85.65 | 95.08 | **88.77** | **89.14** | 95.23 | **96.52** | 92.52 |
| SND | **96.00** | **97.67** | 96.43 | **87.08** | **98.36** | 87.55 | 88.73 | **96.82** | 96.38 | **93.89** |

In Table 4, which evaluates AUPRC on the same dataset, SND again demonstrates robust performance with an average AUPRC of 96.53%. It achieves high results in key classes such as class 0 (98.12%) and class 7 (98.79%), outperforming GNL's average of 94.19%. ERM and IRM show competitive, but lower results, underscoring SND's superior capability in novelty detection under domain shifts.

In Table 5, we summarize the average AUROC results for novelty detection using the Kurcuma dataset, which includes seven distinct scenarios: SYNTHETIC, AKUD, CLIPART, EKUD, EKUD-M1, EKUD-M2, and EKUD-M3. Each scenario corresponds to a specific category, 0 for bottle opener, 1 for can opener, 2 for fork, 3 for knife, 4 for pizza cutter, 5 for spatula, 6 for spoon, 7 for tongs, and 8 for whisk. SND achieves the best average AUROC of 69.96%, outperforming other methods across most categories, including 0 (72.70%) and 4 (78.67%). GNL shows strong results in category 2 (77.32%) but falls short overall with an average of 61.08%. ERM and IRM trail behind with averages of 50.12% and 49.64%.

Table 6 presents the average AUPRC scores. SND again leads with an average of 93.89%, excelling in categories like 0 (96.00%) and 4 (98.36%). GNL performs well with an average of 92.52%, while ERM and IRM show moderate performance, averaging around 89%. These results highlight the superior performance of SND across varying domain shifts.

In conclusion, the combined analysis of AUROC and AUPRC metrics highlights SND's strengths in novelty detection. Its strong performance in both metrics places it ahead of existing techniques, showing great potential for future research and practical applications

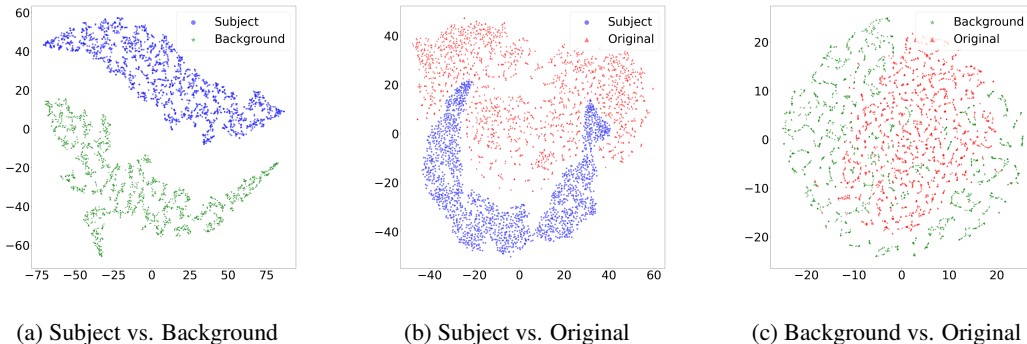

(a) Subject vs. Background     (b) Subject vs. Original     (c) Background vs. Original

Figure 3: t-SNE visualizations illustrating the separation of features: (a) Subject vs. Background, (b) Subject vs. Original, and (c) Background vs. Original. We choose the class of "0" in Multi-background MNIST to provide the visualization result.

**T-SNE Analysis and Visualization**

In addition, we employed image visualizations, t-SNE analysis, and quantitative evaluation to demonstrate how our approach enhances novelty detection performance under domain shifts.

Figures 3a, 3b and 3c demonstrate the model's capacity to isolate subject features across varying scenarios, as shown through t-SNE visualizations. Specifically, Figure 3a shows the t-SNE projection of subject and background features, revealing distinct clusters that highlight effective feature separation.

Figure 3b highlights the t-SNE results for subject and original image features, further confirming that the model retains essential subject information while discarding irrelevant background details. Finally, in 3c, the comparison of background features with those of the original images reveals the model's capacity to distinguish between background elements and the overall image characteristics.

The t-SNE plots collectively support the model's effectiveness in handling domain shifts, underscoring the importance of the subject in achieving high accuracy in novelty detection. Additional experimental results are provided in the Appendix B for further analysis.

## 5 CONCLUSION

In this paper, we propose a novel approach to novelty detection (ND) named SND. This method disentangles subject and background information across different scenes and detects novelties using only subject features. By reducing the mutual information between subject and background, we achieve effective separation, demonstrating that our model significantly outperforms existing methods in ND scenarios with domain shifts. Experimental results demonstrate the method's exceptional performance in novelty detection scenarios where the testing data distribution differs from the training data. The proposed SND offers new insights and methods for ND research, holding significant importance for real-world novelty detection tasks.

Future work could further optimize the SND method and explore its performance on more complex datasets and practical applications to validate its broad applicability and robustness. We anticipate that SND will play a greater role in high-stakes domains (such as finance, healthcare, and defence intelligence), helping achieve more reliable novelty detection.

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

APPENDIX FOR SND

## A  MI ESTIMATION

Mutual information (MI) is a fundamental measure of dependence between two random variables. From an information-theoretic perspective, when learning distinct latent embeddings $\mathbf{z}_s$ and $\mathbf{z}_b$, it is preferable to minimize the mutual information between them. When $\mathbf{z}_s$ and $\mathbf{z}_b$ are independent, we can directly obtain the feature vectors of the subject and the background respectively. The mutual information between the subject part $\mathbf{z}_s$ and the background part $\mathbf{z}_b$ is defined as:

$$I(\mathbf{z}_s; \mathbf{z}_b) = \mathbb{E}_{p(\mathbf{z}_s, \mathbf{z}_b)} \left[ \log \frac{p(\mathbf{z}_s, \mathbf{z}_b)}{p(\mathbf{z}_s)p(\mathbf{z}_b)} \right] \tag{12}$$

With feature pairs $\{(\mathbf{z}_s^i, \mathbf{z}_b^i)\}_{i=1}^N$, the mutual information $I(\mathbf{z}_s; \mathbf{z}_b)$ can be estimated as:

$$\hat{I}_{\text{MI}} = \frac{1}{N} \sum_{i=1}^N \log p(\mathbf{z}_b^i | \mathbf{z}_s^i) - \frac{1}{N^2} \sum_{i=1}^N \sum_{j=1}^N \log p(\mathbf{z}_b^j | \mathbf{z}_s^i)$$

$$= \frac{1}{N^2} \sum_{i=1}^N \sum_{j=1}^N \left[ \log p(\mathbf{z}_b^i | \mathbf{z}_s^i) - \log p(\mathbf{z}_b^j | \mathbf{z}_s^i) \right] \tag{13}$$

In the estimation $\hat{I}_{\text{MI}}$, $\log p(\mathbf{z}_b^i | \mathbf{z}_s^i)$ represents the conditional log-likelihood of the subject pair $(\mathbf{z}_s^i, \mathbf{z}_b^i)$, and $\{\log p(\mathbf{z}_b^j | \mathbf{z}_s^i)\}_{j=1}^N$ represents the conditional log-likelihood of the background information for the pair $(\mathbf{z}_s^i, \mathbf{z}_b^j)$. The difference between $\log p(\mathbf{z}_b^i | \mathbf{z}_s^i)$ and $\log p(\mathbf{z}_b^j | \mathbf{z}_s^i)$ is the contrastive log-ratio between the two conditional distributions.

When the conditional distribution $p(\mathbf{z}_b | \mathbf{z}_s)$ is known, MI can be directly estimated using equation 13 with samples $\{(\mathbf{z}_s^i, \mathbf{z}_b^i)\}_{i=1}^N$.

However, in our experiments, calculating MI according to the above method is challenging because the relationship between subject and background variables is unknown. To solve this problem, we approximate $p(\mathbf{z}_b | \mathbf{z}_s)$ using a variational distribution $\xi_{\theta_m}(\mathbf{z}_b | \mathbf{z}_s)$ with parameters $\theta_m$. Given this setup, the mutual information between subject and background can be expressed as:

$$I_{\text{MI}}(\mathbf{z}_s; \mathbf{z}_b) = \mathbb{E}_{p(\mathbf{z}_s, \mathbf{z}_b)}[\log \xi_{\theta_m}(\mathbf{z}_b | \mathbf{z}_s)] - \mathbb{E}_{p(\mathbf{z}_s)} \mathbb{E}_{p(\mathbf{z}_b)}[\log \xi_{\theta_m}(\mathbf{z}_b | \mathbf{z}_s)] \tag{14}$$

Similar to the MI estimator $\hat{I}_{\text{MI}}$ in equation 13, the unbiased estimator for MI with samples $\{(\mathbf{z}_s^i, \mathbf{z}_b^i)\}_{i=1}^N$ is:

$$\hat{I}_{\text{MI}} = \frac{1}{N^2} \sum_{i=1}^N \sum_{j=1}^N \left[ \log \xi_{\theta_m}(\mathbf{z}_b^i | \mathbf{z}_s^i) - \log \xi_{\theta_m}(\mathbf{z}_b^j | \mathbf{z}_s^i) \right] \tag{15}$$

$$= \frac{1}{N} \sum_{i=1}^N \left[ \log \xi_{\theta_m}(\mathbf{z}_b^i | \mathbf{z}_s^i) - \frac{1}{N} \sum_{j=1}^N \log \xi_{\theta_m}(\mathbf{z}_b^j | \mathbf{z}_s^i) \right] \tag{16}$$

According to Cheng et al. (2020), using the variational approximation $\xi_{\theta_m}$, the modified MI no longer guarantees an upper bound for $I(x; y)$. However, the modified MI shares good properties with the original MI. With a good variational approximation $\xi_{\theta_m}$, the modified MI can still hold an upper bound on mutual information.

## B  ABLATION EXPERIMENTS

We analyze the AUROC performance of anomaly detection models trained on features extracted from three categories: subject, background, and original images. The dataset used for this analysis

consists of images with three backgrounds Multi-background MNIST dataset for the training set and a separate green background for the test set. The results in Figure 4 show that subject features have the most significant impact on model performance, with the AUROC scores exhibiting substantial variation across different classes. For certain classes, the subject feature AUROC approaches 90, indicating its strong discriminative power. In contrast, background features demonstrate consistently lower AUROC scores, suggesting a limited contribution to distinguishing anomalies, while original image features show intermediate performance.

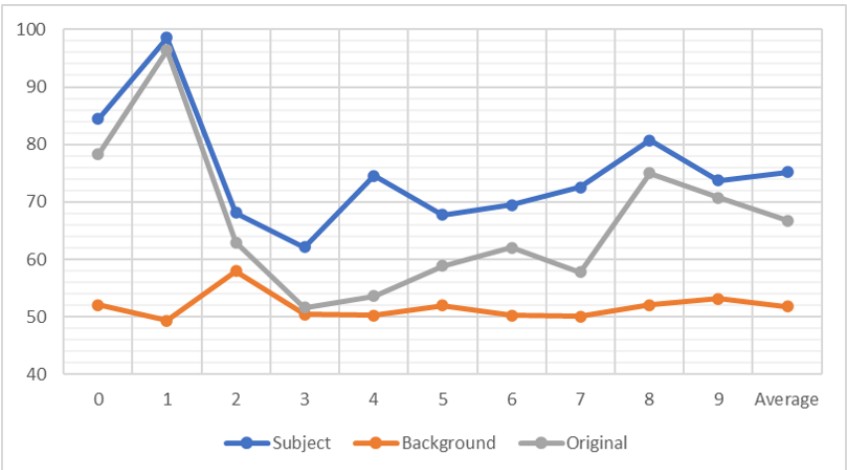

Figure 4: AUROC performance comparison across different image features (Subject, Background, and Original) for novelty detection.

## C ALGORITHM OVERVIEW

In Algorithm 1, we present the flowchart illustrating our method. This process is divided into two main stages: the extraction of subject and background information, followed by novelty detection. The algorithm begins by initializing the network parameters and performing feature extraction, decomposition, and mutual information minimization to ensure statistical independence between subject and background features. To model the background components, a deep Gaussian Mixture Model (GMM) is employed, and the overall loss function is computed for optimization. The testing stage focuses on computing the novelty score for each test sample, which is used for identifying novel subjects.

## D MODEL STRUCTURE

In our proposed anomaly detection model, we integrate a Variational Autoencoder (VAE), a Contrastive Mutual Information Upper Bound (CLUB) module, and a Gaussian Mixture Model (GMM) to address complex background and subject separation. The Encoder consists of four convolutional layers with 64, 128, 256, and 512 filters, respectively. Each convolutional layer has a kernel size of 4x4, a stride of 2, and padding of 1, followed by batch normalization and LeakyReLU activation. The output is then flattened and passed through fully connected layers to generate 128-dimensional latent vectors for both the background and the subject.

The Decoder reconstructs the input image using transposed convolutional layers with the same structure as the encoder, but in reverse. Specifically, the decoder has four layers with 512, 256, 128, and 64 filters, and similarly employs batch normalization and LeakyReLU activation. The final output layer uses a Sigmoid activation function to produce the reconstructed image.

To ensure effective disentanglement between the background and subject features, the CLUB module estimates mutual information by learning mean and log variance through two fully connected networks. Each network consists of linear layers with 128 inputs, followed by a 64-unit hidden layer, LeakyReLU activation, and dropout.

---

**Algorithm 1** Subject Information Extraction for Novelty Detection with Domain Shifts (SND)

---

**Require:** Training dataset $\mathcal{D} = \{\mathbf{x}_1, \mathbf{x}_2, \ldots, \mathbf{x}_N\}$, the number of different backgrounds $K$, testing dataset $\mathcal{D}' = \{\mathbf{x}'_i\}_{i=1}^M$.

1: **Stage 1: Extracting subject and background information**
2: Initialize the network parameters $\{\theta_f, \theta_s, \theta_b, \theta_m, \theta_g, \theta'_s, \theta'_b\}$
3: **for** each training epoch **do**
4:     **for** each randomly sampled minibatch $\{\mathbf{x}_i\}_{i=t_1}^{t_L}, \{t_1, b_2, \ldots, t_L\} \subseteq \{1, 2, \ldots, N\}$ **do**
5:         Encode the data by $\mathbf{z}_f^i = G_{\theta_f}(\mathbf{x}_i)$.
6:         Decompose $\mathbf{z}_f^i$ into subject features $\mathbf{z}_s^i$ and background features $\mathbf{z}_b^i$ as equation 4
7:         Minimize mutual information (MI) between $\mathbf{z}_s^i$ and $\mathbf{z}_b^i$ to ensure separation as equation 3
8:         Fit a deep Gaussian Mixture Model (GMM) to $\mathbf{z}_b^i$ to identify background components as equation 6
9:         Compute the total loss $L_{\text{total}}$ as described in equation 10
10:         Update $\{\theta_f, \theta_s, \theta_b, \theta_m, \theta_g, \theta'_s, \theta'_b\}$ through back-propagation.
11:     **end for**
12: **end for**
13: **Stage 2: Novelty detection**
14: **for** each test sample $\mathbf{x}_{\text{new}}$ in $\mathcal{D}'$ **do**
15:     Extract $\mathbf{z}_s^{\text{new}} = F_{\theta_s}(G_{\theta_f}(\mathbf{x}_{\text{new}}))$
16:     Compute the **novelty score** based on the extracted $\mathbf{z}_s^{\text{new}}$ and $\mathcal{D}_s$ as equation 11
17: **end for**
18: **Output:** Predictions $\{\hat{f}(\mathbf{x}_i)\}_{i=1}^n$

---

Additionally, the background latent space is modelled using a Gaussian Mixture Model (GMM) with three components. The GMM estimates the mean and covariance of the background latent vectors, which are used to compute energy-based novelty scores, helping the model identify outliers based on background variations.

# E  DATASET DESCRIPTION

**Multi-background MNIST and Multi-background Fashion-MNIST Datasets**

The Colored MNIST dataset was originally developed by IRM (Arjovsky et al., 2019) to encourage classifiers to overfit on spurious features such as colour, rather than focusing on the intrinsic shape features of the digits. For our specific task, we expanded on this concept by creating the Multi-background MNIST and Multi-background Fashion-MNIST (Fashion-MNIST) datasets. These are based on the original MNIST and Fashion-MNIST datasets but introduce significant domain shifts to further challenge the models' generalization capabilities.

The training set consists of images of digits (0-9) displayed on backgrounds with four different colours: yellow, purple, red, and blue. The testing set consists of digits (0-9) placed on a green background, which was not seen during training. Each digit is treated as the normal class in turn, with the remaining digits considered anomalies for both training and testing.

In our study, we modified the original MNIST dataset, which features white digits on a black background. We randomly selected 4,000 images for the training set and 1,000 images for the test set. In the modified dataset, we replaced the white digits with red, and the black backgrounds were sequentially changed to various colours. In the first variation, the training set images have backgrounds in white, purple, and blue, while the test set images feature a green background. In the second variation, the training set backgrounds include yellow, white, purple, and blue, with the test set still featuring a green background.

As we can see from Figure 5a, the training set consists of images of digits (0-9) displayed on backgrounds with four different colours: yellow, purple, red, and blue. The testing set consists of digits (0-9) placed on a green background, which was not seen during training. Each digit is treated as the normal class in turn, with the remaining digits considered anomalies for both training and testing.

Figure 5: Visualization of the Multi-background MNIST dataset(a) and Multi-background Fashion-MNIST dataset(b).

For the Fashion-MNIST dataset, we applied a similar approach. In the first variation, the training set images have backgrounds in white, green, and blue, and the test set images have a yellow background. In the second variation, the training set backgrounds include green, white, purple, and blue with the test set still featuring a yellow background. These modifications were designed to test the model's ability to generalize across different background colours and domain shifts.

As we can see from Figure 5b, the training set of Fashion-MNIST consists of images of fashion items (such as T-shirts, trousers, shoes, etc.) displayed on four different background colours: blue, green, purple, and white. The testing set consists of fashion items placed on the yellow background, which were not seen during training. Each category of fashion item is treated as the normal class in turn, with the remaining categories considered anomalies for both training and testing.

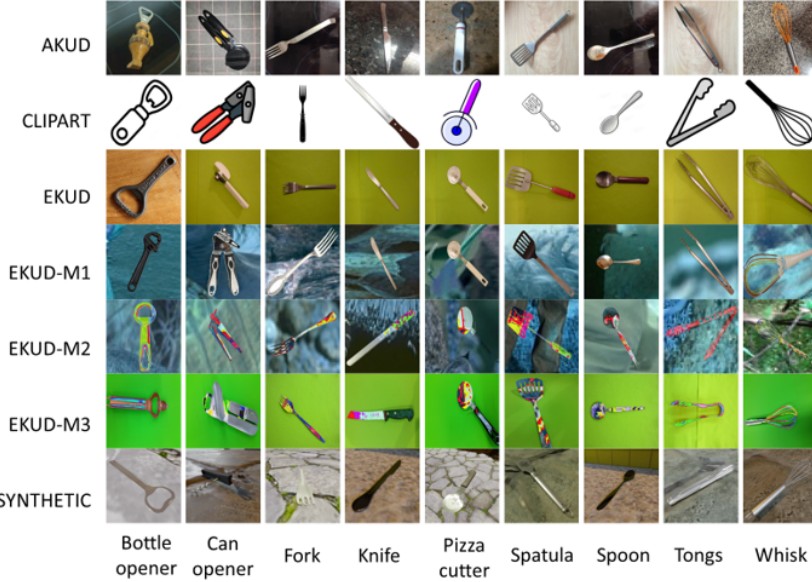

Figure 6: Visualization of the Kurcuma dataset. Each row represents a different dataset corpus, with varying backgrounds ranging from real-world scenes (AKUD, EKUD) to synthetic (SYNTHETIC) and clipart representations. The columns represent different kitchen utensil categories used for the classification task. Each category was treated as the normal class, while all others were considered anomalies.

**Kurcuma**

The Kurcuma collection is a comprehensive dataset for kitchen utensil recognition, specifically targeting domain adaptation (DA) research in robotic home-assistance scenarios, which is shown in Figure 6. It comprises seven distinct corpora, including four developed by the authors, featuring colour images across nine classes: bottle opener, can opener, fork, knife, pizza cutter, spatula, spoon,

tongs, and whisk. The images are captured in various scenes, including uniform backgrounds, textured surfaces, cluttered environments, and synthetic and clipart representations, with each image having a consistent resolution of 256x256 pixels. This dataset is labelled, making it suitable for supervised, unsupervised, and semi-supervised learning tasks.

In our experiments, each background was treated as unseen during training, and we performed a complete training and testing cycle for each. Specifically, we designated one category as the normal class for each background, while the remaining categories were treated as anomalies. This setup allowed us to evaluate the robustness of our model to background shifts and class imbalances across various kitchen utensil types.

A key component of this collection is the Edinburgh Kitchen Utensil Database (EKUD), which includes 897 real-world images of utensils against uniform backgrounds. Following a curation process, where we merged similar classes and eliminated under-represented or low-quality images, the dataset was refined to 618 images across the nine classes. Additionally, the EKUD-M1 corpus modifies the backgrounds of EKUD images using patches from the Berkeley Segmentation Data Set, creating a diverse set of 600 images that enrich the dataset for effective domain adaptation research. Each image in these corpora is annotated with details such as class labels, background type, and image source, facilitating further analysis and experimentation.

This setup highlights the challenges posed by background shifts and the need for models capable of performing well in unseen environments, critical for domain adaptation tasks in real-world applications.

# F    RESULTS AND DISCUSSION ON MULTI-BACKGROUND MNIST

In this study, we evaluated several novelty detection methods on the Multi-background MNIST dataset with a green background as the test set, where the training dataset consists of four different backgrounds, using two metrics AUROC and AUPRC. The focus of our analysis is on the performance of our method, SND, which is highlighted in the last row of each table. By comparing SND with other established methods, we aim to demonstrate its effectiveness in accurately identifying novel instances across diverse backgrounds.

In Table 7, the analysis of AUROC results for the Multi-background MNIST dataset shows that the proposed SND method achieves the best overall performance with an average AUROC of 72.57%. SND significantly outperforms other methods, particularly on digits like 1 (98.02%) and 7 (86.41%). Comparatively, domain adaptation techniques such as ERM and IRM, which achieved averages of 51.03% and 50.12%, respectively, struggled to maintain high accuracy under domain shifts. Other methods like GNL performed well on some specific digits, such as 3 (71.71%), but still fell short in terms of overall consistency, further highlighting SND's robust generalization capabilities across unseen backgrounds.

Table 7: Average AUROCs (%) in novelty detection on Multi-background MNIST with four different coloured backgrounds used in the training set and the unseen green background used in the testing set. In each case, the best result is marked in bold.

| Method | 0 | 1 | 2 | 3 | 4 | 5 | 6 | 7 | 8 | 9 | Average |
|---|---|---|---|---|---|---|---|---|---|---|---|
| COPOD | 62.82 | 70.33 | 63.77 | 64.41 | 65.80 | 64.33 | 64.44 | 66.05 | 63.81 | 65.95 | 65.17 |
| SUOD | 64.52 | 67.42 | 65.79 | 67.72 | 70.17 | 69.37 | 65.46 | 67.24 | 65.10 | 67.70 | 67.05 |
| MO_GAAL | 61.41 | 72.20 | 69.40 | 77.73 | 65.74 | 58.00 | 71.48 | 71.23 | 73.13 | 73.63 | 69.40 |
| DeepSVDD | 63.92 | 58.84 | 72.46 | 56.69 | 48.24 | 88.63 | 66.82 | 76.00 | 62.02 | 62.95 | 65.66 |
| ALAD | 27.97 | 29.07 | 7.58 | 17.22 | 8.85 | 25.06 | 13.84 | 9.77 | 22.37 | 16.93 | 17.87 |
| ECOD | 57.29 | 61.46 | 60.37 | 60.41 | 62.13 | 61.19 | 60.22 | 61.29 | 59.67 | 61.53 | 60.56 |
| INNE | 61.23 | 57.83 | 63.72 | 63.15 | 61.80 | 66.64 | 62.72 | 65.09 | 58.84 | 61.37 | 62.24 |
| AnoGAN | 59.06 | 70.27 | 75.75 | 27.77 | 87.28 | 80.19 | 72.81 | 54.83 | 50.95 | 73.60 | 65.25 |
| ERM | 56.68 | 53.57 | 51.27 | 50.75 | 46.47 | 47.39 | 54.16 | 42.23 | 58.90 | 48.90 | 51.03 |
| IRM | 37.24 | 95.95 | 42.65 | 42.42 | 40.03 | 42.11 | 47.41 | 60.75 | 48.88 | 43.75 | 50.12 |
| GNL | 30.42 | 95.83 | 50.18 | **71.71** | 61.38 | 66.02 | 64.33 | 69.82 | 44.78 | 68.54 | 62.30 |
| SND | **61.47** | **98.02** | **66.21** | 67.38 | **68.93** | **68.73** | **70.77** | **86.41** | **67.74** | **70.05** | **72.57** |

Table 8, the AUPRC results similarly emphasize SND's superior performance with an average score of 95.61%. The method's success in detecting novelty is particularly evident on digits like 1 (99.73%) and 7 (97.78%). In comparison, models like GNL (93.46%) showed competitive but lower results, and traditional methods such as COPOD and SUOD lagged significantly behind, with averages around 24%. ERM and IRM performed better in terms of AUPRC, both near 90%, but were still outperformed by SND across all digits. These results collectively demonstrate the effectiveness of SND in addressing domain shifts and enhancing novelty detection in complex environments.

Table 8: Average AUPRCs (%) in novelty detection on Multi-background MNIST with four different colored backgrounds used in the training set and the unseen green background used in the testing set. In each case, the best result is marked in bold.

| Method | 0 | 1 | 2 | 3 | 4 | 5 | 6 | 7 | 8 | 9 | Average |
|---|---|---|---|---|---|---|---|---|---|---|---|
| COPOD | 22.78 | 27.06 | 23.24 | 23.56 | 24.28 | 23.51 | 23.57 | 24.41 | 23.27 | 24.36 | 24.00 |
| SUOD | 23.37 | 24.93 | 24.22 | 25.21 | 26.93 | 26.31 | 23.99 | 24.93 | 23.79 | 25.30 | 24.90 |
| MO_GAAL | 28.55 | 35.31 | 35.44 | 38.52 | 29.82 | 28.87 | 37.22 | 32.62 | 35.99 | 35.87 | 33.82 |
| DeepSVDD | 26.82 | 22.42 | 43.75 | 23.85 | 25.15 | 67.47 | 31.67 | 37.81 | 41.87 | 30.85 | 35.17 |
| ALAD | 13.95 | 14.02 | 11.36 | 12.64 | 11.53 | 16.20 | 12.05 | 11.68 | 13.51 | 12.75 | 12.97 |
| ECOD | 20.47 | 22.11 | 21.64 | 21.66 | 22.41 | 22.01 | 21.61 | 22.04 | 21.38 | 22.15 | 21.75 |
| INNE | 24.32 | 23.76 | 26.23 | 25.97 | 25.81 | 27.78 | 25.75 | 27.13 | 24.26 | 24.82 | 25.58 |
| AnoGAN | 13.94 | 13.72 | 21.56 | 25.21 | 25.04 | 21.62 | 14.73 | 22.03 | 16.68 | 22.23 | 19.68 |
| ERM | 91.46 | 90.00 | 90.63 | 90.06 | 89.72 | 90.27 | 90.96 | 87.29 | 92.35 | 89.92 | 90.26 |
| IRM | 86.87 | 99.47 | 88.93 | 88.17 | 88.42 | 89.44 | 89.75 | 92.55 | 90.26 | 88.80 | 90.27 |
| GNL | 86.40 | 99.41 | 91.91 | **95.52** | 93.61 | 94.73 | 93.62 | 94.88 | 89.81 | 94.75 | 93.46 |
| SND | **93.23** | **99.73** | **94.51** | 94.48 | **95.21** | **95.49** | **95.61** | **97.78** | **95.17** | **94.89** | **95.61** |

In conclusion, the results show that SND is the most effective method for novelty detection on the Multi-background MNIST dataset. Its superiority in both AUROC and AUPRC, combined with its consistent performance across various categories, highlights its adaptability and precision. These findings suggest that SND is a highly reliable and robust solution for novelty detection, outperforming other approaches in both accuracy and precision.

# G    RESULTS AND DISCUSSION ON MULTI-BACKGROUND FASHION-MNIST

In this study, we conducted experiments on the Multi-background Fashion-MNIST dataset with a yellow background as the test set, where the training dataset consists of three different backgrounds.

Table 9: Average AUROCs (%) in novelty detection on Multi-background Fashion-MNIST. The best result is marked in bold.

| Method | T-shirt | Trou-ser | Pull-over | Dress | Coat | Sandal | Shirt | Sneaker | Bag | Ankle boot | Average |
|---|---|---|---|---|---|---|---|---|---|---|---|
| COPOD | 50.58 | 49.40 | 50.37 | 49.62 | 50.28 | 50.01 | 51.06 | 49.92 | 54.64 | 50.07 | 50.59 |
| SUOD | 51.14 | 48.33 | 51.57 | 50.96 | 51.97 | 50.36 | 53.45 | 48.18 | 57.20 | 51.74 | 51.49 |
| MO_GAAL | 41.62 | 30.33 | 82.63 | 25.56 | 46.63 | 33.89 | 63.64 | 19.00 | 46.02 | 56.31 | 44.56 |
| DeepSVDD | 53.58 | 29.64 | 45.44 | 47.80 | 42.56 | 55.28 | 46.83 | 53.32 | 44.02 | 35.68 | 45.42 |
| ALAD | 53.37 | 49.22 | 38.78 | 60.01 | 44.76 | 59.90 | 46.44 | 50.96 | 46.86 | 58.50 | 50.88 |
| ECOD | 49.61 | 46.99 | 48.99 | 47.42 | 48.83 | 49.87 | 50.48 | 48.26 | 54.83 | 49.86 | 49.51 |
| INNE | 56.15 | 52.63 | 57.98 | 54.68 | 58.02 | 65.53 | 59.93 | 58.08 | 63.30 | 59.97 | 58.63 |
| AnoGAN | 46.01 | 4.20 | 79.02 | 18.28 | 55.58 | 53.98 | 62.57 | 47.46 | **66.36** | 31.66 | 46.51 |
| ERM | 57.30 | 36.19 | 56.52 | 43.69 | 47.72 | 90.98 | 50.58 | 38.41 | 56.44 | 34.58 | 51.24 |
| IRM | 57.19 | 30.25 | 58.02 | 43.18 | 44.73 | 91.05 | 52.51 | 41.28 | 57.53 | 32.09 | 50.78 |
| GNL | 69.95 | **97.70** | 37.43 | **82.85** | 50.00 | 83.46 | 54.25 | 90.27 | 53.51 | 31.06 | 65.05 |
| SND | **79.36** | 93.36 | **84.58** | 74.46 | **75.33** | **91.29** | **77.93** | **91.92** | 65.49 | **94.39** | **82.81** |

In Table 9,, the AUROC results for novelty detection on the Multi-background Fashion-MNIST dataset show that SND outperforms other methods with an average AUROC of 82.81%. SND

achieves strong results across various classes, particularly for T-shirt (79.36%), Ankle boot (94.39%), and Sneaker (91.92%). This demonstrates the model's capability to generalize across different object types. Comparatively, GNL performs well on Trouser (97.70%) and Sneaker (90.27%) but falls short in other categories, resulting in a lower overall average of 65.05%. ERM and IRM also lag behind with averages of 51.24

In Table 10, SND again leads in AUPRC with an average of 97.62%, surpassing GNL's 94.02%. SND excels in categories such as Ankle boot (99.34%) and Sneaker (99.08%), highlighting its robustness in novelty detection. ERM and IRM, while competitive in AUPRC with averages around 90%, still fall short of SND's performance. This consistent superiority in both AUROC and AUPRC confirms SND's ability to efficiently detect novelties even under significant domain shifts.

Table 10: Average AUPRCs (%) in novelty detection on Multi-background Fashion-MNIST. The best result is marked in bold.

| Method | T-shirt | Trouser | Pull-over | Dress | Coat | Sandal | Shirt | Sneaker | Bag | Ankle boot | Average |
|---|---|---|---|---|---|---|---|---|---|---|---|
| COPOD | 22.72 | 22.36 | 22.69 | 22.32 | 22.61 | 22.57 | 22.88 | 22.55 | 24.85 | 22.53 | 22.81 |
| SUOD | 23.69 | 21.86 | 23.93 | 22.76 | 23.89 | 22.56 | 24.70 | 21.88 | 26.64 | 23.01 | 23.49 |
| MO_GAAL | 23.05 | 18.51 | 64.89 | 16.82 | 23.36 | 18.67 | 44.11 | 15.73 | 24.76 | 32.16 | 28.21 |
| DeepSVDD | 27.94 | 20.30 | 23.31 | 25.67 | 22.96 | 30.85 | 24.49 | 27.57 | 22.25 | 20.35 | 24.57 |
| ALAD | 26.97 | 26.87 | 19.65 | 28.74 | 22.01 | 30.79 | 22.24 | 28.29 | 22.61 | 31.60 | 25.98 |
| ECOD | 22.89 | 21.53 | 22.78 | 21.52 | 22.55 | 22.48 | 23.23 | 22.01 | 25.36 | 22.34 | 22.67 |
| INNE | 28.38 | 26.86 | 29.16 | 28.07 | 29.22 | 36.27 | 30.36 | 31.00 | 31.82 | 30.38 | 30.15 |
| AnoGAN | 24.82 | 13.93 | 52.54 | 16.16 | 32.49 | 24.28 | 41.05 | 22.49 | 35.82 | 18.27 | 28.18 |
| ERM | 93.83 | 85.09 | 93.41 | 88.30 | 91.97 | 98.13 | 91.75 | 85.13 | 92.33 | 85.32 | 90.53 |
| IRM | 93.87 | 83.41 | 93.61 | 88.22 | 90.81 | 97.99 | 92.67 | 86.30 | 92.38 | 84.45 | 90.37 |
| GNL | 95.59 | **99.72** | 88.87 | **97.94** | 91.35 | 98.02 | 92.54 | 98.95 | 91.30 | 85.91 | 94.02 |
| SND | **96.69** | 98.95 | **97.88** | 96.65 | **96.78** | 98.87 | 97.08 | 99.08 | 94.84 | 99.34 | 97.62 |

## H    RESULTS AND DISCUSSION ON KURCUMA

In the subsequent experiments, we selected different backgrounds to serve as anomalous settings for detection. To clearly present the results, we used numerical labels to represent each category: 0 corresponds to bottle opener, 1 to can opener, 2 to fork, 3 to knife, 4 to pizza cutter, 5 to spatula, 6 to spoon, 7 to tongs, and 8 to whisk. The results from the tables illustrate a comparative analysis of different novelty detection methods on the Kurcuma dataset across multiple environments, including SYNTHETIC, AKUD, CLIPART, EKUD, and its variations (M1, M2 and M3). Each table presents the performance of various methods in terms of average AUROC and AUPRC, offering a clear view of the strengths and weaknesses of these approaches in different settings.

When considering the AUROC results, our method, SND, consistently outperforms the others across all environments. For instance, in the synthetic environment (Table 11), SND achieves the highest average AUROC of 65.28%, with strong performance in categories like 2 (73.62%) and 4 (71.84%). In the AKUD environment (Table 13), SND again leads with an average AUROC of 65.02%, significantly surpassing other methods such as COPOD and DeepSVDD. This pattern continues in the other environments (Table 15,Table 17,Table 19,Table 21,Table 23), where SND delivers the highest AUROC, demonstrating its robustness across varied backgrounds.

In terms of AUPRC (Table12, Table14, Table 16, Table18, Table20, Table22, Table24), SND consistently achieves near-perfect precision, further solidifying its effectiveness in novelty detection. For example, in the synthetic environment, SND obtains an average AUPRC of 93.57%, outperforming all other methods in nearly every category. Similarly, in the AKUD and clipart environments, SND leads with AUPRC values of 92.05% and 94.81%, respectively. These results indicate that SND not only excels at detecting anomalies but also maintains high precision in classifying true positive instances.

In summary, the analysis of both AUROC and AUPRC metrics across different environments highlights SND as the most effective method for novelty detection on the Kurcuma dataset. Its con-

sistently high performance in varied and challenging scenarios demonstrates its adaptability and reliability, making it a superior choice for tasks requiring accurate and precise anomaly detection. These findings emphasize the potential of SND for future applications in novelty detection across diverse domains.

Table 11: Average AUROCs (%) in novelty detection on the Kurcuma dataset using data from the SYNTHETIC environment as the test set.

| Method | 0 | 1 | 2 | 3 | 4 | 5 | 6 | 7 | 8 | Average |
|--------|------|------|------|------|------|------|------|------|------|---------|
| ALAD | 45.86 | 45.87 | 57.29 | 56.37 | 57.20 | 44.43 | 51.43 | 54.81 | 50.30 | 51.51 |
| COPOD | 48.73 | 50.40 | 48.82 | 47.93 | 55.89 | 57.17 | 52.02 | 44.60 | 48.84 | 50.49 |
| DeepSVDD | 52.99 | 44.59 | 50.62 | 55.77 | 48.00 | 49.38 | 39.53 | 49.13 | 50.71 | 48.97 |
| ECON | 50.89 | 46.92 | 49.68 | 51.39 | 55.01 | 59.29 | 45.22 | 43.63 | 48.13 | 50.02 |
| INNE | 52.47 | 43.70 | 51.13 | 54.49 | 52.56 | 53.82 | 42.98 | 44.22 | 53.41 | 49.86 |
| AnoGAN | 50.86 | 45.45 | 51.05 | 50.97 | 53.35 | 58.60 | 41.89 | 46.56 | 48.37 | 49.68 |
| ERM | 51.01 | 47.32 | 49.64 | 53.28 | 46.21 | 42.78 | 51.02 | 54.64 | 52.75 | 49.85 |
| IRM | 49.91 | 46.65 | 50.08 | 51.35 | 46.71 | 42.81 | 51.54 | 55.05 | 52.29 | 49.60 |
| GNL | 45.13 | 39.63 | 63.97 | 60.34 | 54.75 | **64.86** | 60.54 | 54.28 | **68.55** | 56.89 |
| SND | **60.49** | **65.11** | **73.62** | **64.36** | **71.84** | 58.97 | **66.91** | **58.66** | 67.52 | **65.28** |

Table 12: Average AUPRCs (%) in novelty detection on the Kurcuma dataset using data from the SYNTHETIC environment as the test set.

| Method | 0 | 1 | 2 | 3 | 4 | 5 | 6 | 7 | 8 | Average |
|--------|------|------|------|------|------|------|------|------|------|---------|
| ALAD | 89.51 | 89.29 | 92.28 | 75.38 | 93.33 | 89.41 | 91.38 | 91.90 | 91.07 | 89.28 |
| COPOD | 89.83 | 92.14 | 90.41 | 72.03 | 92.60 | 93.51 | 91.74 | 89.91 | 89.42 | 89.07 |
| DeepSVDD | 91.00 | 89.73 | 91.16 | 75.79 | 91.67 | 91.26 | 89.12 | 89.67 | 91.19 | 88.95 |
| ECON | 90.43 | 90.96 | 90.94 | 74.72 | 92.47 | 92.62 | 89.69 | 88.99 | 90.22 | 89.00 |
| INNE | 90.98 | 90.22 | 91.75 | 75.15 | 92.11 | 91.21 | 89.32 | 90.03 | 90.99 | 89.08 |
| AnoGAN | 90.60 | 90.56 | 91.23 | 73.65 | 92.37 | **93.63** | 88.94 | 89.22 | 90.75 | 88.99 |
| ERM | 91.25 | 90.09 | 90.55 | 75.20 | 90.27 | 88.78 | 91.51 | 92.07 | 90.50 | 88.91 |
| IRM | 90.75 | 90.19 | 90.31 | 73.98 | 90.92 | 88.77 | 91.81 | 92.21 | 91.03 | 88.89 |
| GNL | 88.73 | 89.81 | 94.35 | 76.66 | 92.71 | 92.94 | 92.25 | 92.71 | **94.43** | 90.51 |
| SND | **93.17** | **95.00** | **96.22** | **85.74** | **96.23** | 93.03 | **95.68** | **93.56** | 93.49 | **93.57** |

Table 13: Average AUROCs (%) in novelty detection on the Kurcuma dataset using data from the AKUD environment as the test set.

| Method | 0 | 1 | 2 | 3 | 4 | 5 | 6 | 7 | 8 | Average |
|--------|------|------|------|------|------|------|------|------|------|---------|
| ALAD | 52.12 | 47.03 | 52.18 | 53.66 | 49.89 | 46.12 | 58.95 | 46.45 | 50.03 | 50.71 |
| COPOD | 49.67 | 28.30 | 63.17 | 51.98 | 45.88 | 41.13 | 53.09 | 40.97 | 46.33 | 46.72 |
| DeepSVDD | 42.33 | 32.59 | 61.60 | 48.24 | 43.65 | 32.59 | 56.53 | 44.84 | 53.01 | 46.15 |
| ECON | 47.01 | 27.34 | 63.93 | 52.19 | 43.43 | 38.97 | 61.83 | 40.39 | 51.00 | 47.34 |
| INNE | 44.64 | 21.61 | 69.22 | 47.41 | 44.22 | 40.49 | 66.69 | 47.20 | 62.84 | 49.37 |
| AnoGAN | 45.62 | 43.71 | 54.79 | 53.05 | 52.76 | 41.51 | 40.44 | 40.67 | 44.25 | 46.31 |
| ERM | 51.12 | 50.42 | 48.17 | 46.01 | 59.35 | 48.42 | 42.51 | 60.56 | 55.72 | 51.37 |
| IRM | 50.85 | 49.30 | 47.92 | 46.50 | 61.68 | 57.98 | 43.92 | 60.04 | 55.20 | 52.60 |
| GNL | 40.09 | 28.27 | **75.32** | **67.15** | 43.62 | 53.01 | **79.29** | 45.85 | 64.06 | 55.18 |
| SND | **59.55** | **63.25** | 73.12 | 55.80 | **69.32** | **61.17** | 58.54 | **72.22** | **72.24** | **65.02** |

Table 14: Average AUPRCs (%) in novelty detection on the Kurcuma dataset using data from the AKUD environment as the test set.

| Method | 0 | 1 | 2 | 3 | 4 | 5 | 6 | 7 | 8 | Average |
|---|---|---|---|---|---|---|---|---|---|---|
| ALAD | 84.95 | 93.53 | 94.35 | 66.63 | 93.14 | 92.95 | 89.10 | 94.15 | 95.06 | 89.32 |
| COPOD | 85.95 | 92.20 | 95.80 | 66.79 | 93.22 | 91.67 | 86.78 | 92.76 | 93.99 | 88.80 |
| DeepSVDD | 84.20 | 92.38 | 95.49 | 63.55 | 92.93 | 90.83 | 87.35 | 93.88 | 94.71 | 88.37 |
| ECON | 84.91 | 92.10 | 95.58 | 67.26 | 93.00 | 91.34 | 89.03 | 92.89 | 94.54 | 88.96 |
| INNE | 84.63 | 90.58 | 96.98 | 61.04 | 92.49 | 92.50 | 92.53 | 94.60 | 97.00 | 89.15 |
| AnoGAN | 83.64 | 94.08 | 94.43 | 67.72 | 93.14 | 90.78 | 82.58 | 92.94 | 94.46 | 88.20 |
| ERM | 85.02 | 95.05 | 93.47 | 63.45 | 94.75 | 93.50 | 82.26 | 95.97 | 95.31 | 88.75 |
| IRM | 84.88 | 95.09 | 92.80 | 62.65 | 95.42 | 95.15 | 82.80 | 95.92 | 95.20 | 88.88 |
| GNL | 83.31 | 92.65 | **97.70** | **73.29** | 91.36 | 95.01 | **95.31** | 94.34 | 96.41 | 91.04 |
| SND | **88.28** | **97.10** | 97.46 | 71.33 | **96.32** | 95.50 | 88.28 | **97.09** | **97.05** | **92.05** |

Table 15: Average AUROCs (%) in novelty detection on the Kurcuma dataset using data from the CLIPART environment as the test set.

| Method | 0 | 1 | 2 | 3 | 4 | 5 | 6 | 7 | 8 | Average |
|---|---|---|---|---|---|---|---|---|---|---|
| ALAD | 44.27 | 64.30 | 42.34 | 65.56 | 45.90 | 55.27 | 58.66 | 55.56 | 57.28 | 54.35 |
| COPOD | 28.33 | 34.57 | 79.49 | 61.72 | 22.67 | 50.47 | 57.33 | 46.35 | 46.95 | 47.54 |
| DeepSVDD | 44.69 | 46.28 | 38.59 | 58.55 | 50.99 | 51.38 | 54.27 | 64.78 | 50.06 | 51.07 |
| ECON | 30.25 | 35.42 | 76.90 | 59.23 | 23.00 | 50.20 | 57.65 | 48.48 | 47.74 | 47.65 |
| INNE | 40.27 | 50.03 | 64.05 | 63.27 | 25.47 | 50.91 | 54.44 | 54.00 | 48.22 | 50.07 |
| AnoGAN | 67.28 | 68.59 | 19.47 | 50.66 | **74.08** | 49.48 | 47.38 | 57.01 | 61.74 | 55.08 |
| ERM | 68.47 | 61.95 | 27.16 | 40.59 | 73.63 | 53.95 | 44.38 | 54.89 | 35.14 | 51.13 |
| IRM | 74.20 | 64.59 | 26.10 | 43.45 | 70.30 | 45.41 | 47.25 | 55.66 | 37.39 | 51.59 |
| GNL | 45.38 | 40.42 | 76.14 | 44.79 | 38.23 | **67.30** | **80.86** | 48.82 | **69.48** | 56.82 |
| SND | **76.56** | **69.40** | **83.51** | **74.94** | 67.43 | 55.94 | 70.46 | **70.14** | 65.50 | **70.43** |

Table 16: Average AUPRCs (%) in novelty detection on the Kurcuma dataset using data from the CLIPART environment as the test set.

| Method | 0 | 1 | 2 | 3 | 4 | 5 | 6 | 7 | 8 | Average |
|---|---|---|---|---|---|---|---|---|---|---|
| ALAD | 85.88 | 95.45 | 84.32 | 94.65 | 92.23 | 83.94 | 87.28 | 93.68 | 94.82 | 90.25 |
| COPOD | 81.43 | 90.94 | 95.77 | 94.01 | 85.98 | 80.98 | 87.76 | 92.11 | 93.71 | 89.19 |
| DeepSVDD | 85.16 | 91.88 | 83.81 | 93.48 | 91.73 | 81.79 | 87.51 | 95.77 | 93.08 | 89.36 |
| ECON | 81.85 | 91.11 | 95.24 | 93.62 | 85.97 | 80.78 | 87.69 | 92.41 | 93.74 | 89.16 |
| INNE | 84.34 | 93.36 | 92.31 | 93.71 | 87.01 | 80.85 | 86.85 | 93.01 | 93.65 | 89.45 |
| AnoGAN | 92.37 | **96.74** | 75.75 | 91.62 | 97.19 | 81.52 | 84.02 | 93.63 | 95.88 | 89.86 |
| ERM | 92.69 | 95.72 | 77.87 | 87.61 | 96.69 | 82.57 | 82.12 | 93.93 | 90.49 | 88.85 |
| IRM | 94.34 | 95.83 | 77.89 | 89.52 | 95.71 | 79.43 | 83.47 | 94.18 | 91.38 | 89.08 |
| GNL | 85.76 | 91.95 | 95.36 | 89.22 | 90.99 | **90.05** | **94.41** | 92.10 | **96.57** | 91.82 |
| SND | **95.27** | 96.03 | **97.17** | **96.35** | **97.63** | 85.93 | 92.02 | **96.54** | 96.32 | **94.81** |

Table 17: Average AUROCs (%) in novelty detection on the Kurcuma dataset using data from the EKUD environment as the test set.

| Method | 0 | 1 | 2 | 3 | 4 | 5 | 6 | 7 | 8 | Average |
|---|---|---|---|---|---|---|---|---|---|---|
| ALAD | 41.48 | 31.14 | 47.02 | 55.28 | 35.22 | 44.77 | 35.96 | 47.91 | 43.25 | 42.45 |
| COPOD | 43.18 | 42.46 | 47.27 | 47.59 | 60.86 | 49.65 | 59.40 | 44.11 | 56.52 | 50.12 |
| DeepSVDD | 42.89 | 44.16 | 49.93 | 47.45 | 55.63 | 59.78 | 61.82 | 48.60 | 55.75 | 51.78 |
| ECON | 43.78 | 41.07 | 47.18 | 45.69 | 57.71 | 52.68 | 59.78 | 42.29 | 50.53 | 48.97 |
| INNE | 41.40 | 34.75 | 62.20 | 51.81 | 44.61 | 36.19 | 72.95 | 35.82 | 46.72 | 47.38 |
| AnoGAN | 36.12 | 50.25 | 50.58 | 49.72 | 60.83 | 42.36 | 52.47 | 35.29 | 46.41 | 47.11 |
| ERM | 59.41 | 40.04 | 52.20 | 53.96 | 47.09 | 43.84 | 41.55 | 60.54 | 48.65 | 49.70 |
| IRM | 50.98 | 49.82 | 55.69 | 46.69 | 54.09 | 44.53 | 63.05 | 51.26 | 0.00 | 46.23 |
| GNL | 45.87 | 48.98 | **89.62** | **86.01** | 58.71 | **72.86** | **91.78** | 73.38 | 76.48 | 71.52 |
| SND | **84.27** | **75.31** | 73.05 | 68.58 | **87.41** | 67.62 | 66.95 | **89.78** | **85.37** | **77.59** |

Table 18: Average AUPRCs (%) in novelty detection on the Kurcuma dataset using data from the EKUD environment as the test set.

| Method | 0 | 1 | 2 | 3 | 4 | 5 | 6 | 7 | 8 | Average |
|---|---|---|---|---|---|---|---|---|---|---|
| ALAD | 94.09 | 94.05 | 90.74 | 82.77 | 96.49 | 75.38 | 68.17 | 92.87 | 91.24 | 87.31 |
| COPOD | 95.22 | 95.84 | 89.71 | 78.52 | 98.56 | 80.49 | 77.96 | 93.66 | 94.01 | 89.33 |
| DeepSVDD | 94.88 | 95.60 | 90.80 | 79.23 | 98.35 | 82.94 | 80.35 | 94.16 | 94.36 | 90.07 |
| ECON | 95.67 | 95.58 | 88.95 | 77.59 | 98.46 | 82.06 | 77.36 | 93.52 | 93.81 | 89.22 |
| INNE | 94.98 | 94.18 | 93.44 | 82.23 | 97.71 | 74.39 | 84.00 | 92.57 | 93.54 | 89.67 |
| AnoGAN | 94.17 | 96.12 | 90.68 | 79.36 | 98.63 | 78.29 | 77.84 | 92.28 | 93.42 | 88.98 |
| ERM | 96.85 | 95.46 | 92.11 | 81.57 | 97.68 | 74.27 | 68.84 | 95.53 | 92.78 | 88.34 |
| IRM | 96.55 | 96.35 | 91.18 | 83.47 | 97.59 | 82.06 | 69.60 | 96.12 | 92.54 | 89.50 |
| GNL | 94.78 | 96.75 | **98.70** | **96.41** | 98.40 | **89.09** | **97.14** | 97.66 | 96.67 | 96.18 |
| SND | **99.03** | **98.92** | 95.35 | 88.17 | **99.65** | 86.59 | 84.80 | 97.54 | **99.22** | **98.78** |

Table 19: Average AUROCs (%) in novelty detection on the Kurcuma dataset using data from the EKUD-M1 environment as the test set.

| Method | 0 | 1 | 2 | 3 | 4 | 5 | 6 | 7 | 8 | Average |
|---|---|---|---|---|---|---|---|---|---|---|
| ALAD | 45.10 | 50.03 | 51.66 | 52.45 | 49.34 | 51.97 | 50.07 | 51.77 | 45.09 | 49.72 |
| COPOD | 53.50 | 43.83 | 52.41 | 51.39 | 45.69 | 45.84 | 49.62 | 57.27 | 54.66 | 50.47 |
| DeepSVDD | 58.20 | 48.75 | 49.74 | 51.37 | 43.65 | 43.35 | 49.72 | 57.74 | 44.43 | 49.66 |
| ECON | 56.00 | 45.49 | 50.96 | 52.76 | 41.80 | 44.72 | 50.63 | 59.33 | 46.92 | 49.85 |
| INNE | 54.34 | 42.71 | 50.36 | 53.80 | 54.16 | 38.73 | 53.95 | 57.86 | 58.30 | 51.58 |
| AnoGAN | 53.09 | 52.74 | 54.08 | 51.55 | 46.03 | 46.86 | 49.72 | 54.01 | 47.37 | 50.61 |
| ERM | 54.83 | 47.75 | 50.07 | 50.54 | 54.51 | 47.10 | 51.78 | 47.10 | 55.09 | 50.97 |
| IRM | 49.66 | 47.34 | 50.03 | 48.28 | 55.35 | 50.53 | 51.41 | 47.81 | 54.36 | 50.53 |
| GNL | 44.72 | 45.21 | 68.45 | **86.97** | 56.24 | **63.08** | 61.64 | **74.25** | **74.85** | 63.93 |
| SND | **73.84** | **74.57** | **72.07** | 64.82 | **82.84** | 58.60 | **62.53** | 67.06 | 67.97 | **69.37** |

Table 20: Average AUPRCs (%) in novelty detection on the Kurcuma dataset using data from the EKUD-M1 environment as the test set.

| Method | 0 | 1 | 2 | 3 | 4 | 5 | 6 | 7 | 8 | Average |
|---|---|---|---|---|---|---|---|---|---|---|
| ALAD | 94.09 | 94.05 | 90.74 | 82.77 | 96.49 | 75.38 | 68.17 | 92.87 | 91.24 | 87.31 |
| COPOD | 95.22 | 95.84 | 89.71 | 78.52 | 98.56 | 80.49 | 77.96 | 93.66 | 94.01 | 89.33 |
| DeepSVDD | 94.88 | 95.60 | 90.80 | 79.23 | 98.35 | 82.94 | 80.35 | 94.16 | 94.36 | 90.07 |
| ECON | 95.67 | 95.58 | 88.95 | 77.59 | 98.46 | 82.06 | 77.36 | 93.52 | 93.81 | 89.22 |
| INNE | 94.98 | 94.18 | 93.44 | 82.23 | 97.71 | 74.39 | 84.00 | 92.57 | 93.54 | 89.67 |
| AnoGAN | 94.17 | 96.12 | 90.68 | 79.36 | 98.63 | 78.29 | 77.84 | 92.28 | 93.42 | 88.98 |
| ERM | 96.71 | 95.48 | 90.25 | 82.04 | 97.79 | 77.33 | 73.68 | 94.56 | 94.28 | 89.12 |
| IRM | 96.08 | 96.15 | 90.82 | 79.97 | 97.91 | 79.37 | 73.03 | 93.78 | 93.33 | 88.94 |
| GNL | 94.78 | 96.75 | **98.70** | **96.41** | 98.40 | **89.09** | **97.14** | **97.66** | 96.67 | 96.18 |
| SND | **99.03** | **98.92** | 95.35 | 88.17 | **99.65** | 86.59 | 84.80 | 97.54 | **99.22** | **98.78** |

Table 21: Average AUROCs (%) in novelty detection on the Kurcuma dataset using data from the EKUD-M2 environment as the test set.

| Method | 0 | 1 | 2 | 3 | 4 | 5 | 6 | 7 | 8 | Average |
|---|---|---|---|---|---|---|---|---|---|---|
| ALAD | 49.12 | 51.28 | 44.00 | 53.53 | 54.99 | 53.97 | 48.50 | 49.08 | 56.57 | 51.23 |
| COPOD | 50.71 | 63.20 | 49.74 | 47.39 | 66.30 | 43.94 | 50.75 | 52.15 | 52.29 | 52.94 |
| DeepSVDD | 46.71 | 63.69 | 50.96 | 46.19 | 59.33 | 46.02 | 47.28 | 54.27 | 50.73 | 51.69 |
| ECON | 49.29 | 69.29 | 51.88 | 48.44 | 66.47 | 44.85 | 47.86 | 54.20 | 51.36 | 53.74 |
| INNE | 48.75 | 61.52 | 50.39 | 49.74 | 57.63 | 44.62 | 50.61 | 54.17 | 56.22 | 52.63 |
| AnoGAN | 41.77 | 65.88 | 50.51 | 48.79 | 65.74 | 45.48 | 51.09 | 56.53 | 52.56 | 53.15 |
| ERM | 48.06 | 35.68 | 48.81 | 54.30 | 34.21 | 53.36 | 51.51 | 49.88 | 49.66 | 47.27 |
| IRM | 47.58 | 32.85 | 50.59 | 54.35 | 33.93 | 52.43 | 51.01 | 48.16 | 48.26 | 46.57 |
| GNL | 58.57 | 42.00 | **83.79** | **67.45** | 50.86 | 56.94 | **69.78** | 62.93 | **70.46** | 62.53 |
| SND | **80.34** | **80.87** | 69.11 | 65.96 | **85.78** | **57.51** | 59.26 | **79.08** | 69.38 | **71.92** |

Table 22: Average AUPRCs (%) in novelty detection on the Kurcuma dataset using data from the EKUD-M2 environment as the test set.

| Method | 0 | 1 | 2 | 3 | 4 | 5 | 6 | 7 | 8 | Average |
|---|---|---|---|---|---|---|---|---|---|---|
| ALAD | 95.83 | 96.50 | 88.91 | 82.50 | 97.79 | 81.29 | 74.06 | 94.50 | 94.18 | 89.51 |
| COPOD | 95.40 | 98.02 | 90.30 | 81.19 | 98.85 | 74.70 | 74.18 | 95.28 | 93.61 | 89.06 |
| DeepSVDD | 95.95 | 97.73 | 90.36 | 81.37 | 98.44 | 77.57 | 72.41 | 94.87 | 91.94 | 88.96 |
| ECON | 95.40 | 98.27 | 90.90 | 80.32 | 98.90 | 77.10 | 72.78 | 94.98 | 93.25 | 89.10 |
| INNE | 95.33 | 97.11 | 90.30 | 81.56 | 98.19 | 76.36 | 73.16 | 94.69 | 94.51 | 89.02 |
| AnoGAN | 94.98 | 98.07 | 90.42 | 82.08 | 98.71 | 75.28 | 74.26 | 95.31 | 93.51 | 89.18 |
| ERM | 95.44 | 94.99 | 89.33 | 83.71 | 96.52 | 80.67 | 73.88 | 94.00 | 93.39 | 89.10 |
| IRM | 95.58 | 94.51 | 90.55 | 83.45 | 96.08 | 80.40 | 73.59 | 93.78 | 92.54 | 88.94 |
| GNL | 97.31 | 95.43 | **97.64** | 87.49 | 97.33 | **84.24** | 86.38 | 95.73 | **96.59** | 93.13 |
| SND | **99.07** | **99.18** | 95.21 | **89.24** | **99.61** | 82.88 | **98.32** | **99.41** | 96.61 | **95.50** |

Table 23: Average AUROCs (%) in novelty detection on the Kurcuma dataset using data from the EKUD-M3 environment as the test set.

| Method | 0 | 1 | 2 | 3 | 4 | 5 | 6 | 7 | 8 | Average |
|---|---|---|---|---|---|---|---|---|---|---|
| ALAD | 55.23 | 39.13 | 46.94 | 50.28 | 39.00 | 49.35 | 38.54 | 45.24 | 49.37 | 45.90 |
| COPOD | 44.40 | 44.17 | 46.72 | 47.30 | 59.62 | 50.07 | 59.17 | 44.78 | 56.65 | 50.32 |
| DeepSVDD | 53.72 | 42.81 | 46.04 | 48.52 | 57.68 | 55.70 | 55.73 | 34.51 | 46.84 | 49.06 |
| ECON | 44.59 | 42.14 | 46.04 | 44.68 | 56.31 | 54.44 | 58.93 | 43.26 | 50.97 | 49.04 |
| INNE | 47.77 | 36.34 | 63.27 | 53.74 | 39.71 | 35.48 | **71.40** | 39.93 | 44.38 | 48.00 |
| AnoGAN | 37.02 | 42.28 | 50.15 | 48.58 | 52.44 | 48.84 | 62.92 | 37.07 | 41.08 | 46.71 |
| ERM | 54.67 | 38.52 | 53.31 | 54.25 | 43.17 | 49.09 | 46.78 | 60.86 | 54.22 | 50.54 |
| IRM | 52.79 | 43.41 | 50.20 | 60.73 | 41.24 | 47.60 | 43.59 | 56.19 | 57.43 | 50.35 |
| GNL | 65.26 | 45.71 | **83.93** | 47.53 | 39.29 | 58.21 | 59.89 | **68.29** | 77.82 | 60.66 |
| SND | **73.84** | **72.38** | 74.43 | **66.47** | **86.09** | **60.08** | 64.62 | 65.13 | 67.97 | **70.11** |

Table 24: Average AUPRCs (%) in novelty detection on the Kurcuma dataset using data from the EKUD-M3 environment as the test set.

| Method | 0 | 1 | 2 | 3 | 4 | 5 | 6 | 7 | 8 | Average |
|---|---|---|---|---|---|---|---|---|---|---|
| ALAD | 96.92 | 95.24 | 90.60 | 81.81 | 96.93 | 76.29 | 67.75 | 92.28 | 93.32 | 87.91 |
| COPOD | 95.26 | 96.00 | 89.27 | 78.20 | 98.46 | 80.70 | 78.01 | 93.71 | 94.01 | 89.29 |
| DeepSVDD | 96.53 | 95.19 | 89.26 | 78.62 | 98.58 | 82.00 | 78.42 | 91.93 | 93.37 | 89.32 |
| ECON | 95.73 | 95.89 | 88.56 | 77.23 | 98.37 | 82.62 | 77.12 | 93.61 | 93.84 | 89.22 |
| INNE | 95.69 | 94.89 | 93.29 | 82.76 | 97.48 | 73.19 | **83.94** | 92.98 | 92.96 | 89.69 |
| AnoGAN | 94.26 | 95.47 | 89.62 | 78.63 | 97.88 | 79.34 | 81.05 | 92.54 | 91.93 | 88.97 |
| ERM | 95.91 | 94.85 | 92.31 | 82.10 | 97.33 | 77.43 | 73.38 | 96.27 | 93.73 | 89.26 |
| IRM | 95.95 | 95.44 | 89.40 | 86.62 | 97.02 | 77.36 | 70.56 | 94.95 | 94.67 | 89.11 |
| GNL | 97.61 | 95.09 | **97.84** | 79.86 | 96.98 | 84.27 | 81.51 | 96.54 | **97.61** | 91.92 |
| SND | **98.70** | **98.60** | 97.10 | **90.02** | **99.58** | **84.79** | 81.91 | **96.84** | 95.61 | **93.68** |