# OpenReview forum: "Subject Information Extraction for Novelty Detection with Domain Shifts"
_ICLR.cc/2025/Conference — Submitted to ICLR 2025_

### Official Review · Reviewer_Ygvq · 2024-10-30

**Soundness:** 3
**Presentation:** 3
**Contribution:** 2
**Rating:** 5
**Confidence:** 3

**Summary:**

This paper addresses Unsupervised Novelty Detection (UND) under domain shifts, where training and testing data differ due to background conditions. By separating core subject information from domain-specific background variations, the proposed method uses only subject representations for novelty detection, enhancing performance under domain shifts. Experiments show generalization to new domains, outperforming baseline methods.

**Strengths:**

- The paper is well-written and easy to understand.
- The setting considered in the paper is interesting.
- Extensive experiments validate the claims and demonstrate the superior performance of the proposed method.

**Weaknesses:**

- The scenario where training and testing data differ solely due to background conditions may not be entirely practical. I suggest the authors provide additional examples to illustrate realistic applications of this setting.

- The authors reference related works (Oza et al., 2020; Yang et al., 2023; Carvalho et al., 2024) that have already explored anomaly detection under domain shift. Please clarify how the proposed approach differs from these methods and, if possible, provide comparative evaluations.

**Questions:**

Please see Weaknesses

---

### Official Review · Reviewer_Qm8j · 2024-10-31

**Soundness:** 2
**Presentation:** 2
**Contribution:** 1
**Rating:** 3
**Confidence:** 4

**Summary:**

This paper tackles the unsupervised anomaly detection under domain shift problem. Specifically, it introduces a Subject-Novelty Detection (SND) algorithm that disentangles content (or “subject”) information from background in images, allowing to perform unsupervised anomaly detection only on the content part of the image representation. The number of different backgrounds in the training set is assumed to be known in SND. It is validated on two toy datasets (MNIST and Fashion-MNIST) and one small-scale dataset (Kurkuma).

**Strengths:**

• Addressing ouf-of-domain (OOD) generalization for Anomaly Detection (AD) is an interesting and currently understudied topic;

• The idea of separating content from style (or “subject” form “background”) to perform AD is rather straightforward and yet it appears to be new in the literature.

**Weaknesses:**

• SND is a rather restricted algorithm as it only handles OOD generalization for images with new backgrounds. Other settings should be also studied (such as robustness to data corruption or new style [1, 2]) in the context of AD under domain shift. Additionally, the number of backgrounds is assumed to be known in the current algorithm, which is a rather strong assumption compared to other algorithms that do not make it [1, 3, 4].

• Following my previous concern, the benchmarks used in this study are limited as they only reflect a change in the background as a new domain. Other benchmarks – such as PACS, CIFAR10, or MVTec (see [1] or [2], for instance)– should also be included to evaluate the versatility of SND on more challenging domain shifts.

• This study seems to ignore the recent advances in visual AD made possible through large pre-trained (or “foundation”) models such as DINOv2 or CLIP [3, 4]. These models are very efficient zero- or few-shot learners, and they deserve to be evaluated in the current benchmark. This literature is also in line with recent works on disentangling content from style with large pre-trained models [2], which I advise the authors to take inspiration from.

[1] Anomaly Detection under Distribution Shift, Cao et al., ICCV 2023
[2] Simple Disentanglement of Style and Content in Visual Representations, Ngweta, Maity et al., ICML 2023
[3] AnomalyDINO: Boosting Patch-based Few-shot Anomaly Detection with DINOv2, arXiv 2024
[4] WinCLIP: Zero-/Few-Shot Anomaly Classification and Segmentation, CVPR 2023

**Questions:**

• SND introduces several hyper-parameters (weights in the loss, bandwidth in the density estimator), but how these are set in the current experiments is unclear. How did you cross-validate these hyper-parameters?

• The proposed method is evaluated on a real-world dataset (Kurkuma) using only a shallow CNN as the encoder. Did you also consider using large pre-trained models (such as DINOv2) to encode the visual features and perform AD? This should reflect a more realistic setting.

• The ablation study is quite limited: the benefit of imposing the background energy function and the disentanglement term should be evaluated carefully.

---

### Official Review · Reviewer_7rt6 · 2024-11-02

**Soundness:** 2
**Presentation:** 2
**Contribution:** 1
**Rating:** 5
**Confidence:** 3

**Summary:**

This work presents a framework for unsupervised novelty detection problem. It introduces a method that disentangles subject information from background variation encapsulating the domain information to enhance detection performance under domain shifts. It then minimizes the mutual information between the representations of the subject and background while modelling the background variation using a deep Gaussian mixture model, where the novelty detection is conducted on the subject representations nd hence is not affected by the variation of domains. Experiments on MNIST-like datasets demonstrate some effectiveness.

**Strengths:**

* This work tackles a popular problem of unsupervised novelty detection, where domain shift quantification and transfer learning is important in the era of foundation models.
* Experiments demonstrate the effectiveness of the proposed method on some simple baselines, some ablation studies were performed to validate the contribution of certain components.
* Codes and algorithms are available for reproducibility

**Weaknesses:**

* The settings of UND is more commonly addressed as anomaly detection/one-class classification problems, where there are already significant development in this field with many renowned benchmarks (e.g., MVTEC) and methods (e.g., DeepSVDD, PaDIM, CutPaste, etc.). I think the MNIST dataset is the easiest one and while the authors ignored a significant body of competitive baselines and benchmarks.
* The method only separates background from the subjects, which is a coarse-grained feature in images. For instance, a simple OTSU method preinstall in open-cv can isolate the subject from the background, while training on these processed subject image can enable to model to learn sufficient subject information. In this sense, authors should strengthen the motivation and technical contribution of the proposed method.
* Related to above, the experiments are insufficient to meet the bar. Although the authors present a lot of tables, they are mainly on the same datasets (MNIST-like) and only quote different metrics. It is expected that the method to be applied to mode modern datasets (at least TinyImageNet level with 200 classes) to validate its scalability and generalizability to other datasets. Additional metrics are fine but should be presented in the Appendix.
* While checking the codes, the authors only tested their method on simple CNNs, where the commonly used architectures in the field of novelty detection, such as ResNet and ViT, were not tested. This also bring concerns on the scalability or applicability of the proposed method on more modern vision models.
* Although K’ is defined in problem formulations, the authors did not present how they resolve novel background types from the new samples. It seems that the K’ new backgrounds (unseen in any training samples) will still be fitted in the K GMM components, which severely limits its capability on new background types.

Minor:
* Many baselines are of very low performance. This posts questions on whether these baselines are too simple. For instance, DeepSVDD, although a renowned method, is already 6 years old. Authors are suggested to choose more competitive baselines (e.g., the following works of ERM).
* It would be preferred if the standard deviation of each experiment is provided, instead of just reporting the means.

**Questions:**

* The novelty score seems to require a training set to work. How could the method work when we have a pretrained model where we only have access to the training weights?

---

### Official Review · Reviewer_RLDS · 2024-11-03

**Soundness:** 2
**Presentation:** 3
**Contribution:** 2
**Rating:** 3
**Confidence:** 4

**Summary:**

The paper introduces Subject-Novelty Detection (SND), an approach for unsupervised novelty detection in settings with domain shifts between training and testing data. Unlike typical methods, which may misclassify normal samples from unseen domains as novel, SND separates subject information from domain-dependent background variations. By minimizing mutual information between the subject and background representations, SND ensures robust novelty detection across unseen domains, outperforming baseline methods. Comprehensive experiments validate SND's efficacy, showing it achieves state-of-the-art results across challenging datasets such as Multi-background MNIST, Fashion-MNIST, and the Kurcuma dataset.

**Strengths:**

1. The study problem is relevant and important.
2. In general, this paper is well written and easy to follow.

**Weaknesses:**

1. The study of out-of-distribution (OOD) detection under domain shift is a well-established problem in the field. In addition to the related works mentioned in this paper, a simple Google search can yield the following relevant papers. However, the research gaps or limitations of these previous works (including those cited or uncited papers) are not clearly explained.

  a) Bozorgtabar, B., Vray, G., Mahapatra, D., et al. "SOoD: Self-supervised Out-of-Distribution Detection Under Domain Shift for Multi-Class Colorectal Cancer Tissue Types." In Proceedings of the IEEE/CVF International Conference on Computer Vision, 2021: 3324-3333.

   b) Gao, Z., Yan, S., He, X. "ATTA: Anomaly-Aware Test-Time Adaptation for Out-of-Distribution Detection in Segmentation." Advances in Neural Information Processing Systems, 2023, 36: 45150-45171.

2. Similarly, in the experiments, the methods compared are primarily from before 2022, neglecting more recent works in this field.

3. The authors evaluate their experiments solely on the relatively simple MNIST dataset, which seems outdated given current standards. The ability and advantages of the proposed framework in tackling more challenging natural images have not been adequately validated.

4. The proposed solution of separating foreground and background is reasonable and straightforward. However, it remains unclear whether this strategy would still be robust in more complex scenarios, such as when multiple objects exist in an image or when separating the background from the foreground is difficult.

Overall, while this manuscript is generally complete, its contributions and novelty may be incremental, and its advantages to the field are not clearly articulated.

**Questions:**

Please see the above weakness.

---

### Meta-Review · Area_Chair_4ito · 2024-12-17

**Metareview:**

The paper addresses the problem of unsupervised novelty detection under domain shifts, where training and testing data differ due to background variations. To tackle this, the authors propose Subject-Novelty Detection (SND), a method that disentangles subject information from domain-specific background variations. By minimizing mutual information between subject and background representations, SND performs novelty detection solely on the subject representations.

While the paper presents a reasonable and intuitive strategy for disentangling subject and background information, there are notable weaknesses that limit its overall impact. It was noted by all reviewers that the experimental evaluation relies heavily on relatively simple MNIST-based datasets, which do not sufficiently validate the method's robustness or scalability for modern and real-world scenarios. More challenging benchmarks should be explored to demonstrate versatility under broader domain shifts (ie. not only background shift); broader challenges like data corruption, style variations, or pre-trained model use remain unaddressed. The study also does not compare against state-of-the-art architectures/methods (eg. DINOv2, CLIP, etc..), which are widely recognized for achieving superior performance in anomaly detection and domain generalization tasks, although some of these comparisons were started during the rebuttal period -- it was noted that pre-trained models such as AnomalyDINO provided excellent performance compared to SND, without additional training on the target task.

The panel of reviewers believes the paper does not yet meet the standards for ICLR. While the method has merit and introduces an interesting approach to addressing domain shifts, the lack of evaluation on modern datasets and architectures, coupled with strong assumptions and limited comparisons, reduces its overall significance and impact. We encourage the authors to address these concerns in future work by expanding their experiments to more challenging benchmarks, incorporating comparisons with recent state-of-the-art methods, and testing scalability with modern vision architectures. With these improvements, the proposed method could make a stronger contribution to the field of novelty detection and domain generalization.

**Additional Comments On Reviewer Discussion:**

During the rebuttal period, the authors demonstrated started to run a more comprehensive set of comparisons with modern methods, including DINO-type models, and provided additional clarifications on their approach (it was noted that some of the standard pertained models such as DINO were already competitive with the proposed method, unfortunately). While these efforts are appreciated and help address some concerns, the results remain preliminary, and a thorough evaluation on diverse and challenging benchmarks is still needed to fully validate the method's robustness and competitiveness.

---

### Decision · Program_Chairs · 2025-01-22

Reject